# A cross-country analysis of feasible income equality using the sigmoid function and the Boltzmann distribution

Thitithep Sitthiyot[1]*, Kanyarat Holasut[2]

**1** Department of Banking and Finance, Faculty of Commerce and Accountancy, Chulalongkorn University, Bangkok, Thailand, **2** Department of Chemical Engineering, Faculty of Engineering, Khon Kaen University, Khon Kaen, Thailand

* thitithep@cbs.chula.ac.th

## Abstract

This study uses the sigmoid function in combination with the Boltzmann distribution, originally developed by Park and Kim (2021), in order to calculate the optimal income distribution that represents feasible income equality and maximizes total social welfare. Feasible income equality refers to optimal income distribution that is realistically attainable. By employing the data on quintile income shares and the Gini index of 71 countries in 2021 from the World Bank, the results indicate that the optimal income distributions representing feasible income equality, the corresponding values of the Gini index, and the respective shapes of the Lorenz curves of 71 countries are somewhat similar to each other. These results confirm Park and Kim (2021)'s conjecture in that the universal feasible equality line, as depicted by the Lorenz curve, can be identified and applied across multiple countries, potentially serving as a quantitative benchmark. In addition, this study finds that the correlations between the quality of economic and political institutions and the difference between actual and optimal income distributions are negative, suggesting that the better the quality of economic and political institutions is, the closer the gap between actual and optimal income distributions representing feasible income equality. Furthermore, this study estimates the relationship between actual quintile income shares and optimal quintile income shares representing feasible income equality of 71 countries which can be conveniently used to find any approximate level of feasible income share for a particular level of actual income share. Given that high income inequality is associated with health, social, economic, and environmental problems, the overall findings from this study could be useful for designing income redistributive policies and measures.

## Introduction

The gain in income from globalization is not evenly distributed both within a country and across the world and perhaps will not be evenly distributed [1]. Finding an

**Data availability statement:** All relevant data are within the manuscript and its Supporting information files. They are also publicly available and can be accessed from the World Bank websites (https://data.worldbank.org/indicator/ and www.govindicators.org).

**Funding:** The author(s) received no specific funding for this work.

optimal level of income inequality that is beneficial for a country or the world as a whole is thus considered one of major challenges facing scholarly community and policymakers. Studies have shown that high income inequality is associated with health, social, economic, and environmental problems. Subramanian and Kawachi [2] show that high income inequality is strongly associated with rates of infant mortality, heart disease, and several health conditions. Wilkinson and Pickett [3] also report similar findings in that income inequality is correlated with a number of health and social problems, namely, life expectancy, infant mortality, obesity, trust, imprisonment, homicide, drug abuse, mental health, social mobility, childhood education, and teenage pregnancy. Frank et al. [4] find that areas with high income inequality tend to have higher divorce and bankruptcy rate than those with relatively more equal income distributions. As income inequality increases, self-reported happiness diminishes, particularly among income earners who are in the bottom 40% of the income distribution [5]. According to Card et al. [6], when people know their position on the overall income distribution, those with income below the median for their pay unit and occupation report less job satisfaction while those earning above the median report no higher satisfaction. This is because relative disadvantage has a larger negative impact on well-being than relative advantage has a positive impact [7]. In addition to health and social problems, numerous studies have reported the negative relationship between income inequality and economic growth [8–20]. Among these studies, many of them find that the relationship between income inequality and economic growth is nonlinear in that there is an optimal level of income inequality that maximizes economic growth, below which income inequality is conducive to economic growth, and above which income inequality becomes harmful to economic growth [8–10,14,17,18]. Moreover, high income inequality has been shown to exacerbate environmental degradation which, in turn, hinders sustainable economic growth [21–23].

While a number of research have focused on examining the relationships between income inequality and health, social, economic, or environmental factors, as well as finding the optimal level of income inequality that maximizes economic growth as discussed above, to our knowledge, there are few studies, namely, Park et al. [24] and Sitthiyot and Holasut [25], that explore the concept of optimal income distribution representing feasible income equality. This concept was first introduced by Park and Kim [26]. Acknowledging that income equality is idealistic and infeasible in the real world, Park and Kim [26] define optimal income distribution as feasible income equality that not only provides an unbiased allocation of income among different groups of population in a country but also maximizes total social welfare. Given that the sigmoid function has been used in well-being and welfare analysis [27–29] and that the Boltzmann distribution has been applied to the study of income and wealth distributions [30–34], Park and Kim [26] propose using both the sigmoid function and the Boltzmann distribution to calculate the optimal income distribution that represents feasible income equality and maximizes total social welfare.

Regarding the sigmoid function, Park and Kim [26] argue that the sigmoid function is monotonically increasing with a characteristic S-shape that could realistically reflect

a rise in people's welfare as their income increases. When people's income is close to zero, their welfare should be at the minimum level. Below the critical low-income threshold, people's welfare will rise as their income increases but not rapidly since this level of income is still insufficient to provide the basic needs. However, when people's income increases beyond the critical low-income threshold, they begin to have more economic freedom and their welfare will rise more rapidly. As people's income increases further, the degree of economic freedom also rises, but eventually becomes saturated at the critical high-income threshold along with their welfare. Beyond the critical high-income threshold, people's welfare will rise slowly as their income increases.

Concerning the Boltzmann distribution, Park and Kim [26] reason that, in physical sciences, the Boltzmann distribution provides the most probable way that particles in a physical system would be distributed among possible physical substates in thermal equilibrium at a given temperature, which naturally emerges from the maximum entropy principle. The probability of a particle occupying a particular physical substate depends upon its energy and the temperature of that physical system. When applying the Boltzmann distribution to analyze income distribution, Park and Kim [26] argue that a physical system could be replaced by an income distribution system, a physical particle could be replaced by an income unit, a physical substate could be replaced by a group of population, and the potential energy of each physical substate could be replaced by income distribution factor of each group of population. Park and Kim [26] define income distribution factor as a measure of economic contribution that takes various factors such as skills, efforts, and talents into account. Population group with higher income distribution factor should make more contributions and have higher income than population group with lower income distribution factor.

By employing the quintile income share data of four countries with differences in degree of income inequality and socio-economic background, namely, the United States of America (U.S.A), China, Finland, and South Africa, Park and Kim [26] demonstrate, as a proof of principle, that the sigmoid function combined with the Boltzmann distribution can be used to calculate the optimal income distributions that represent feasible income equality and maximize total social welfare for these four countries. Their results, as shown in Table 1, indicate that, in all four countries, the actual income shares of the bottom 20%, the second 20%, and the third 20% are lower than their respective optimal income shares while the actual income shares of the fourth 20% and the top 20% are higher than their respective optimal income shares, except in South Africa where the fourth 20% also receives the actual income share lower than the optimal income share.

The results, as shown in Table 1, also indicate that the optimal quintile income shares of U.S.A., China, Finland, and South Africa are not significantly different from each other, with the bottom 20% ranging between 14.3% (U.S.A.) and

**Table 1. Actual income distributions by quintile and the corresponding values of the Gini index vs. optimal income distributions by quintile representing feasible income equality and the corresponding values of the Gini index of U.S.A., China, Finland, and South Africa.** The results are reproduced from Table 3 in Park and Kim [26] under the Creative Commons Attribution License (CC BY 4.0).

| Country | | Bottom 20% | Second 20% | Third 20% | Fourth 20% | Top 20% | Gini index |
|---|---|---|---|---|---|---|---|
| U.S.A. | Actual | 3.1 | 8.3 | 14.1 | 22.7 | 51.9 | 0.450 |
| | Feasible | 14.3 | 15.6 | 17.3 | 20.0 | 32.8 | 0.170 |
| | Δ | −11.2 | −7.3 | −3.2 | 2.7 | 19.1 | 0.280 |
| China | Actual | 6.5 | 10.7 | 15.3 | 22.2 | 45.3 | 0.360 |
| | Feasible | 15.4 | 16.6 | 17.9 | 20.2 | 29.9 | 0.130 |
| | Δ | −8.9 | −5.9 | −2.6 | 2.0 | 15.4 | 0.230 |
| Finland | Actual | 9.4 | 14.0 | 17.4 | 22.3 | 36.9 | 0.250 |
| | Feasible | 15.5 | 17.1 | 18.5 | 20.6 | 28.4 | 0.120 |
| | Δ | −6.1 | −3.1 | −1.1 | 1.7 | 8.5 | 0.130 |
| South Africa | Actual | 2.4 | 4.8 | 8.2 | 16.5 | 68.2 | 0.570 |
| | Feasible | 15.8 | 16.2 | 16.9 | 18.5 | 32.6 | 0.140 |
| | Δ | −13.4 | −11.4 | −8.7 | −2.0 | 35.6 | 0.430 |

15.8% (South Africa), the second 20% ranging between 15.6% (U.S.A.) and 17.1% (Finland), the third 20% ranging between 16.9% (South Africa) and 18.5% (Finland), the fourth 20% ranging between 18.5% (South Africa) and 20.6% (Finland), and the top 20% ranging between 28.4% (Finland) and 32.8% (U.S.A.). In addition, the values of the Gini index, as a measure of income inequality, corresponding to the optimal income distributions by quintile which represent feasible income equality of these four countries, exhibit a narrow range between 0.120 (Finland) and 0.170 (U.S.A.). According to Park and Kim [26], the similarity of inequality in the optimal income distributions by quintile representing feasible income equality of U.S.A., China, Finland, and South Africa can be shown by the Lorenz curve which depicts the relationship between the cumulative normalized rank of income and the cumulative normalized income as illustrated in Fig 1.

The overall results of optimal income distributions representing feasible income equality of these four countries lead Park and Kim [26] to conjecture that there is "the possibility that a universal feasible equality line could be found and applicable to all countries in the world" which "could be used as a practical guideline for government policies and interventions".

In order to verify Park and Kim [26]'s conjecture as to whether the universal feasible equality line could be identified and applied across multiple countries, this study employs the data on quintile income shares and the Gini index of 71 countries in 2021 from the World Bank [35] and uses the sigmoid function in combination with the Boltzmann distribution, as specified in Park and Kim [26], in order to calculate the optimal income distribution representing feasible income equality for each country. The results from this study could contribute to the existing knowledge on optimal income distribution and, if confirmed, could potentially serve as a quantitative benchmark for designing income redistributive policies and measures.

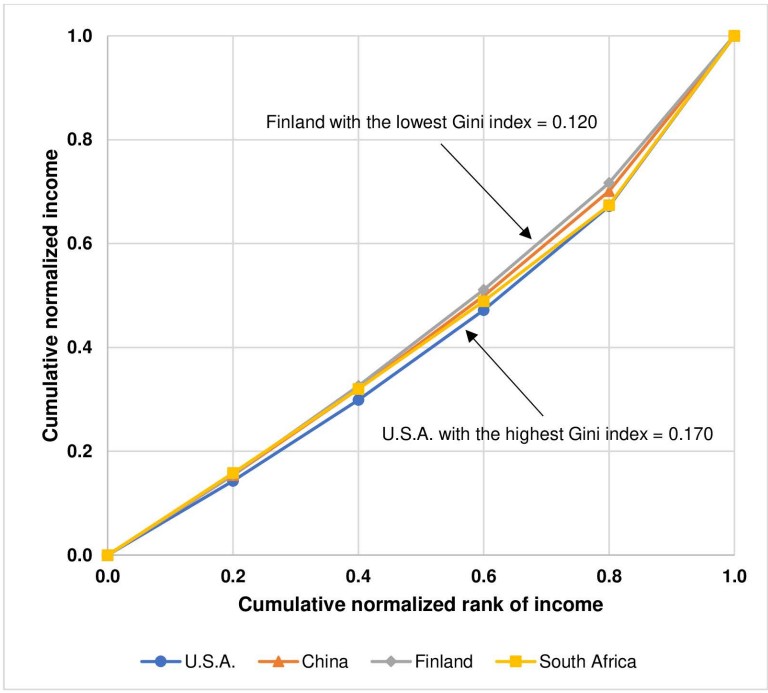

**Fig 1. Lorenz curves depicting the optimal income distributions by quintile representing feasible income equality of U.S.A., China, Finland, and South Africa.** The results are reproduced from Fig 6 in Park and Kim [26] under the Creative Commons Attribution License (CC BY 4.0).

## Materials and methods

This study follows Park and Kim [26]'s method by using the sigmoid function and the Boltzmann distribution in order to calculate optimal income distribution representing feasible income equality and maximizing total social welfare.

For the sigmoid function, let $U$ be the welfare of the quintile population group $i, i = 1, 2, 3, 4, 5$. In addition, let $y_i$ be the quintile income share distributed to the quintile population group i. According to Park and Kim [26], the sigmoid function, representing the welfare of population group i with two parameters which are $\mu$ and $\alpha$, can be specified as shown in Eq 1.

$$U(y_i) = \frac{1}{\left(1 + e^{\alpha*(\mu-y_i)}\right)} \tag{1}$$

Next, let $W(y_1, y_2, y_3, y_4, y_5)$ denote the total social welfare of all quintile population groups. The sigmoid total social welfare function is therefore the sum of $U(y_i)$s as shown in Eq 2.

$$W(y_1, y_2, y_3, y_4, y_5) = \sum_{i=1}^{5} U(y_i) = \sum_{i=1}^{5} \frac{1}{\left(1 + e^{\alpha*(\mu-y_i)}\right)} \tag{2}$$

For the Boltzmann distribution, let $P_i$ be the probability that income is distributed to the quintile population group i. Also, let $Q_i$ be the income distribution factor of the quintile population group i and $\beta$ be a parameter. As discussed in Introduction, $Q_i$ is a measure of economic contribution which, in reality, should be determined by considering various factors such as skills, efforts, and talents. However, to demonstrate the concept as a proof of principle, Park and Kim [26] use income share of quintile population group i as a proxy for $Q_i$ by reasoning that a population group that is in a higher quintile is likely to make more economic contributions and, hence, has more income share than a population group that is in a lower quintile. In the Boltzmann income distribution, $P_i$ can be calculated as shown in Eq 3.

$$P_i = \frac{e^{\beta Q_i}}{\sum_{i=1}^{5} e^{\beta Q_i}}, \quad e = 2.71828 \tag{3}$$

Next, let $Y$ denote the total income that is distributed among quintile groups of population i. Park and Kim [26] set the value of $Y$ to be $100$. Given the value of $Y$, $y_i$, calculated according to the Boltzmann distribution, is expressed as shown in Eq 4.

$$y_i = Y * \frac{e^{\beta Q_i}}{\sum_{i=1}^{5} e^{\beta Q_i}} \tag{4}$$

According to Park and Kim [26], when $y_i$s are inserted into the sigmoid total social welfare function (W), the total social welfare function (W) becomes a function of $\beta$ as shown in Eq 5.

$$\underset{\beta}{\mathrm{Max}} W(y_1, y_2, y_3, y_4, y_5) = \sum_{i=1}^{5} \frac{1}{\left(1 + e^{\alpha*(\mu-y_i)}\right)}, \quad y_i = Y * \frac{e^{\beta Q_i}}{\sum_{i=1}^{5} e^{\beta Q_i}} \tag{5}$$

The parameters μ and α are the critical low-income share threshold and the critical high-income share threshold. Park and Kim [26] define μ as $\frac{(L+H)}{2}$ and α as $\frac{6}{(H-L)}$, where L = $\frac{Q_2+Q_3}{2}$ and H = $\frac{Q_4+Q_5}{2}$, respectively. By taking derivative of Eq 5 with respect to β and solving for the value of β, the sigmoid total social welfare function (W) can be maximized at a specific value of β, denoted as β*, as shown in Eq 6.

$$\frac{\partial W}{\partial \beta} = \frac{\partial W}{\partial y_1} \cdot \frac{\partial y_1}{\partial \beta} + \frac{\partial W}{\partial y_2} \cdot \frac{\partial y_2}{\partial \beta} + \cdots + \frac{\partial W}{\partial y_5} \cdot \frac{\partial y_5}{\partial \beta} = 0$$

(6)

The values of $y_i$s being consistent with β* would then represent the optimal income distribution characterizing feasible income equality and maximizing total social welfare.

This study employs the data on income shares by quintile and the Gini index of 71 countries in 2021 from the World Bank [35] in order to verify Park and Kim [26]'s conjecture as to whether the universal feasible equality line could be identified and applied across multiple countries. The year 2021 is chosen mainly because it is the most recent year with the largest number of countries. These data are publicly available and can be accessed from the World Bank [35].

## Results

This study first reports the results of descriptive statistics of the actual income distributions by quintile and the values of the Gini index of 71 countries. They are shown in Table 2.

The results indicate that there are noticeable differences in the actual income distributions and their inequalities as shown by the minimum and the maximum values of income shares in each quintile as well as the minimum and the maximum values of the Gini index. While the minimum values of income shares of the bottom 20%, the second 20%, the third 20%, the fourth 20%, and the top 20% are equal to 3.1% (Colombia), 7.0% (Colombia), 11.3% (Colombia), 18.8% (Colombia), and 33.3% (Slovak Republic), the maximum values of income shares of the bottom 20%, the second 20%, the third 20%, the fourth 20%, and the top 20% are equal to 10.2% (Slovenia), 15.2% (Slovak Republic), 19.0% (Slovak Republic), 24% (Romania), and 59.8% (Colombia). For the Gini index, the minimum value is 0.241 (Slovak Republic) whereas the maximum value is 0.551 (Colombia). The vast differences in inequality in the actual income distributions by quintile across countries can also be shown by the Lorenz curve. Fig 2 illustrates the Lorenz curves of 71 countries.

Next, this study reports the results of the optimal income distributions by quintile of 71 countries, which represent feasible income equality and maximize total social welfare, calculated using the sigmoid function in conjunction with the Boltzmann distribution. Note that the calculated values of L, H, μ, and α of 71 countries are reported in S1 Table. The results of the optimal income distributions by quintile of 71 countries representing feasible income equality, along with the corresponding values of the Gini index, the associated values of β*, and the maximum values of W ($W_{max}$) are shown in Table 3. Table 3 also reports the actual income distributions by quintile and the corresponding values of the Gini index of 71 countries for the ease of comparison. In addition, the descriptive statistics of the optimal income distributions by quintile

**Table 2. Descriptive statistics of the actual income distributions by quintile and the values of the Gini index of 71 countries.**

|  | Bottom 20% | Second 20% | Third 20% | Fourth 20% | Top 20% | Gini index |
|---|---|---|---|---|---|---|
| **Mean** | 7.2 | 11.8 | 16.1 | 22.1 | 42.8 | 0.353 |
| **Median** | 7.3 | 12.1 | 16.2 | 22.2 | 42.2 | 0.344 |
| **Mode** | 7.6 | 11.0 | 17.3 | 22.5 | 41.5 | 0.329 |
| **Standard Deviation** | 1.6 | 1.7 | 1.5 | 0.9 | 5.3 | 0.065 |
| **Minimum** | 3.1 | 7.0 | 11.3 | 18.8 | 33.3 | 0.241 |
| **Maximum** | 10.2 | 15.2 | 19.0 | 24.0 | 59.8 | 0.551 |

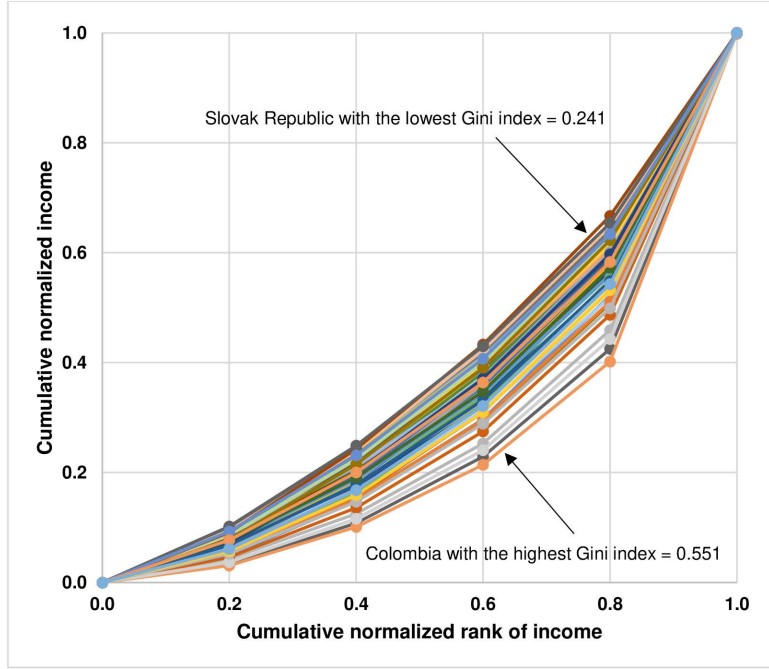

**Fig 2. Lorenz curves depicting inequality in the actual income distributions by quintile of 71 countries.**

representing feasible income equality, along with the corresponding values of the Gini index, $\beta^*$, and $W_{max}$ are reported in Table 4.

The results, as shown in Table 3, indicate that the bottom 20%, the second 20%, and the third 20% receive the actual income shares lower than the optimal income shares in all 71 countries while the fourth 20% and the top 20% in 70 countries receive the actual income shares higher than the optimal income shares. Colombia is the only country where the fourth 20% receives the actual income share lower than the optimal income share. In addition, the values of the Gini index corresponding to the optimal quintile income distributions are lower than those corresponding to the actual quintile income distributions in all 71 countries.

Furthermore, given that the value of $\beta^*$ plays a critical role in maximizing total social welfare ($W_{max}$), this study finds that the values of $\beta^*$ and $W_{max}$ are negatively correlated with the Pearson correlation coefficient (r) = −0.897. The values of $\beta^*$ and $\Delta$ Gini index also show negative correlation with r = −0.846. Note that these results are in line with those reported in Park and Kim (2021). Fig 3 illustrates the scatter plots showing the correlation between the values of $\beta^*$ and $W_{max}$ and that between the values of $\beta^*$ and $\Delta$ Gini index.

In contrast to the results of descriptive statistics of the actual income distributions by quintile and the values of the Gini index of 71 countries, as reported in Table 2, the results of descriptive statistics of the optimal income distributions by quintile representing feasible income equality and the values of the Gini index, as reported in Table 4, indicate that there are similarities in the optimal income distributions and their inequalities across 71 countries, as shown by the narrow ranges between the minimum and the maximum values of income shares in each quintile as well as between the minimum and the maximum values of the Gini index. The minimum values of income shares of the bottom 20%, the second 20%, the third 20%, the fourth 20%, and the top 20% are equal to 13.1% (Slovak Republic), 15.8% (Israel), 17.3% (Colombia), 19.0% (Colombia), and 27.8% (Kazakhstan) whereas the maximum values of income shares of the bottom 20%, the second 20%, the third 20%, the fourth 20%, and the top 20% are equal to 16.3% (Kazakhstan), 17.3%

**Table 3. Actual income distributions by quintile and the corresponding values of the Gini index vs. optimal income distributions by quintile representing feasible income equality and the corresponding values of the Gini index, along with the associated values of $\beta^*$ and $W_{max}$. Total number of countries is 71.**

| Country | | Bottom 20% | Second 20% | Third 20% | Fourth 20% | Top 20% | Gini index |
|---|---|---|---|---|---|---|---|
| Argentina | Actual | 5.0 | 9.8 | 14.9 | 22.5 | 47.8 | 0.424 |
| ($\beta^*$=0.017, $W_{max}$=30.590) | Feasible | 14.9 | 16.2 | 17.6 | 20.1 | 31.2 | 0.147 |
| | Δ | −9.9 | −6.4 | −2.7 | 2.4 | 16.6 | 0.277 |
| Armenia | Actual | 9.4 | 13.8 | 17.3 | 21.8 | 37.7 | 0.279 |
| ($\beta^*$=0.021, $W_{max}$=29.520) | Feasible | 15.7 | 17.2 | 18.5 | 20.3 | 28.3 | 0.113 |
| | Δ | −6.3 | −3.4 | −1.2 | 1.5 | 9.4 | 0.166 |
| Austria | Actual | 7.7 | 13.3 | 17.7 | 22.9 | 38.4 | 0.307 |
| ($\beta^*$=0.024, $W_{max}$=29.122) | Feasible | 14.5 | 16.5 | 18.3 | 20.7 | 29.9 | 0.140 |
| | Δ | −6.8 | −3.2 | −0.6 | 2.2 | 8.5 | 0.167 |
| Belgium | Actual | 9.2 | 14.4 | 17.9 | 22.4 | 36.0 | 0.266 |
| ($\beta^*$=0.024, $W_{max}$=29.102) | Feasible | 15.0 | 17.0 | 18.5 | 20.7 | 28.8 | 0.125 |
| | Δ | −5.8 | −2.6 | −0.6 | 1.7 | 7.2 | 0.141 |
| Benin | Actual | 7.6 | 11.9 | 16.1 | 22.1 | 42.3 | 0.344 |
| ($\beta^*$=0.019, $W_{max}$=30.324) | Feasible | 15.5 | 16.7 | 18.1 | 20.2 | 29.5 | 0.126 |
| | Δ | −7.9 | −4.8 | −2.0 | 1.9 | 12.8 | 0.218 |
| Burkina Faso | Actual | 7.3 | 11.2 | 15.1 | 21.1 | 45.3 | 0.374 |
| ($\beta^*$=0.016, $W_{max}$=30.875) | Feasible | 15.9 | 16.9 | 18.0 | 19.8 | 29.3 | 0.119 |
| | Δ | −8.6 | −5.7 | −2.9 | 1.3 | 16.0 | 0.255 |
| Bulgaria | Actual | 6.1 | 11.0 | 15.5 | 21.9 | 45.6 | 0.390 |
| ($\beta^*$=0.018, $W_{max}$=30.113) | Feasible | 15.2 | 16.5 | 17.9 | 20.0 | 30.4 | 0.136 |
| | Δ | −9.1 | −5.5 | −2.4 | 1.9 | 15.2 | 0.254 |
| Bolivia | Actual | 5.3 | 10.4 | 15.3 | 22.4 | 46.7 | 0.409 |
| ($\beta^*$=0.018, $W_{max}$=30.184) | Feasible | 14.8 | 16.2 | 17.7 | 20.1 | 31.1 | 0.145 |
| | Δ | −9.5 | −5.8 | −2.4 | 2.3 | 15.6 | 0.264 |
| Brazil | Actual | 3.3 | 7.5 | 12.1 | 19.6 | 57.5 | 0.529 |
| ($\beta^*$=0.013, $W_{max}$=31.687) | Feasible | 15.4 | 16.3 | 17.3 | 19.2 | 31.7 | 0.142 |
| | Δ | −12.1 | −8.8 | −5.2 | 0.4 | 25.8 | 0.387 |
| Central African Republic | Actual | 5.6 | 9.7 | 14.1 | 21.4 | 49.2 | 0.430 |
| ($\beta^*$=0.016, $W_{max}$=31.193) | Feasible | 15.5 | 16.5 | 17.7 | 19.8 | 30.5 | 0.133 |
| | Δ | −9.9 | −6.8 | −3.6 | 1.6 | 18.7 | 0.297 |
| China | Actual | 7.4 | 11.4 | 15.7 | 21.9 | 43.6 | 0.357 |
| ($\beta^*$=0.018, $W_{max}$=30.654) | Feasible | 15.6 | 16.7 | 18.0 | 20.1 | 29.5 | 0.125 |
| | Δ | −8.2 | −5.3 | −2.3 | 1.8 | 14.1 | 0.232 |
| Cote d'Ivoire | Actual | 7.6 | 11.6 | 15.7 | 21.8 | 43.3 | 0.353 |
| ($\beta^*$=0.018, $W_{max}$=30.643) | Feasible | 15.7 | 16.8 | 18.1 | 20.1 | 29.3 | 0.123 |
| | Δ | −8.1 | −5.2 | −2.4 | 1.7 | 14.0 | 0.230 |
| Cameroon | Actual | 5.4 | 9.6 | 14.7 | 22.5 | 47.9 | 0.422 |
| ($\beta^*$=0.017, $W_{max}$=30.947) | Feasible | 15.1 | 16.2 | 17.6 | 20.1 | 31.0 | 0.143 |
| | Δ | −9.7 | −6.6 | −2.9 | 2.4 | 16.9 | 0.279 |
| Colombia | Actual | 3.1 | 7.0 | 11.3 | 18.8 | 59.8 | 0.551 |
| ($\beta^*$=0.013, $W_{max}$=32.090) | Feasible | 15.6 | 16.4 | 17.3 | 19.0 | 31.8 | 0.140 |
| | Δ | −12.5 | −9.4 | −6.0 | −0.2 | 28.0 | 0.411 |
| Costa Rica | Actual | 4.3 | 8.2 | 12.8 | 20.6 | 54.1 | 0.487 |
| ($\beta^*$=0.014, $W_{max}$=31.690) | Feasible | 15.5 | 16.4 | 17.5 | 19.5 | 31.2 | 0.138 |

*(Continued)*

| Country | | Bottom 20% | Second 20% | Third 20% | Fourth 20% | Top 20% | Gini index |
|---|---|---|---|---|---|---|---|
| | Δ | −11.2 | −8.2 | −4.7 | 1.1 | 22.9 | 0.349 |
| Cyprus | Actual | 8.6 | 12.8 | 16.8 | 21.8 | 40.1 | 0.313 |
| ($\beta^*$=0.019, $W_{max}$=29.862) | Feasible | 15.6 | 17.0 | 18.3 | 20.2 | 28.8 | 0.118 |
| | Δ | −7.0 | −4.2 | −1.5 | 1.6 | 11.3 | 0.195 |
| Czechia | Actual | 9.7 | 14.4 | 17.5 | 22.0 | 36.3 | 0.262 |
| ($\beta^*$=0.022, $W_{max}$=29.388) | Feasible | 15.6 | 17.3 | 18.5 | 20.5 | 28.2 | 0.114 |
| | Δ | −5.9 | −2.9 | −1.0 | 1.5 | 8.1 | 0.148 |
| Denmark | Actual | 9.2 | 13.9 | 17.3 | 21.8 | 37.8 | 0.283 |
| ($\beta^*$=0.021, $W_{max}$=29.308) | Feasible | 15.6 | 17.2 | 18.5 | 20.3 | 28.4 | 0.115 |
| | Δ | −6.4 | −3.3 | −1.2 | 1.5 | 9.4 | 0.168 |
| Dominican Republic | Actual | 6.6 | 11.1 | 15.2 | 21.5 | 45.7 | 0.385 |
| ($\beta^*$=0.017, $W_{max}$=30.430) | Feasible | 15.5 | 16.7 | 17.9 | 19.9 | 30.0 | 0.129 |
| | Δ | −8.9 | −5.6 | −2.7 | 1.6 | 15.7 | 0.256 |
| Ecuador | Actual | 4.6 | 9.0 | 13.9 | 21.2 | 51.3 | 0.458 |
| ($\beta^*$=0.015, $W_{max}$=31.034) | Feasible | 15.3 | 16.3 | 17.6 | 19.7 | 31.1 | 0.140 |
| | Δ | −10.7 | −7.3 | −3.7 | 1.5 | 20.2 | 0.318 |
| Spain | Actual | 6.4 | 12.3 | 17.2 | 23.6 | 40.4 | 0.339 |
| ($\beta^*$=0.023, $W_{max}$=29.544) | Feasible | 14.0 | 16.1 | 18.0 | 20.9 | 30.9 | 0.155 |
| | Δ | −7.6 | −3.8 | −0.8 | 2.7 | 9.5 | 0.184 |
| Estonia | Actual | 8.1 | 12.3 | 16.7 | 23.1 | 39.9 | 0.318 |
| ($\beta^*$=0.021, $W_{max}$=30.412) | Feasible | 15.0 | 16.5 | 18.1 | 20.7 | 29.7 | 0.134 |
| | Δ | −6.9 | −4.2 | −1.4 | 2.4 | 10.2 | 0.184 |
| Finland | Actual | 9.3 | 13.9 | 17.3 | 22.3 | 37.1 | 0.277 |
| ($\beta^*$=0.022, $W_{max}$=29.730) | Feasible | 15.4 | 17.1 | 18.4 | 20.6 | 28.6 | 0.119 |
| | Δ | −6.1 | −3.2 | −1.1 | 1.7 | 8.5 | 0.158 |
| France | Actual | 7.7 | 13.1 | 17.4 | 22.5 | 39.4 | 0.315 |
| ($\beta^*$=0.022, $W_{max}$=29.072) | Feasible | 14.8 | 16.7 | 18.3 | 20.5 | 29.7 | 0.135 |
| | Δ | −7.1 | −3.6 | −0.9 | 2.0 | 9.7 | 0.180 |
| United Kingdom | Actual | 7.7 | 12.5 | 17.1 | 23.0 | 39.7 | 0.324 |
| ($\beta^*$=0.022, $W_{max}$=29.888) | Feasible | 14.8 | 16.4 | 18.2 | 20.7 | 29.9 | 0.138 |
| | Δ | −7.1 | −3.9 | −1.1 | 2.3 | 9.8 | 0.186 |
| Georgia | Actual | 7.0 | 12.1 | 16.9 | 22.5 | 41.5 | 0.342 |
| ($\beta^*$=0.020, $W_{max}$=29.558) | Feasible | 14.9 | 16.5 | 18.2 | 20.4 | 30.1 | 0.137 |
| | Δ | −7.9 | −4.4 | −1.3 | 2.1 | 11.4 | 0.205 |
| Guinea-Bissau | Actual | 7.9 | 11.9 | 16.2 | 22.4 | 41.6 | 0.334 |
| ($\beta^*$=0.019, $W_{max}$=30.562) | Feasible | 15.5 | 16.7 | 18.1 | 20.4 | 29.4 | 0.126 |
| | Δ | −7.6 | −4.8 | −1.9 | 2.0 | 12.2 | 0.208 |
| Greece | Actual | 7.1 | 12.5 | 17.2 | 23.1 | 40.1 | 0.329 |
| ($\beta^*$=0.022, $W_{max}$=29.519) | Feasible | 14.5 | 16.3 | 18.2 | 20.7 | 30.3 | 0.144 |
| | Δ | −7.4 | −3.8 | −1.0 | 2.4 | 9.8 | 0.185 |
| Croatia | Actual | 8.2 | 13.6 | 17.9 | 23.2 | 37.1 | 0.289 |
| ($\beta^*$=0.025, $W_{max}$=29.420) | Feasible | 14.3 | 16.4 | 18.3 | 21.0 | 29.9 | 0.143 |
| | Δ | −6.1 | −2.8 | −0.4 | 2.2 | 7.2 | 0.146 |
| Hungary | Actual | 9.0 | 13.2 | 17.7 | 22.0 | 38.1 | 0.292 |
| ($\beta^*$=0.021, $W_{max}$=29.536) | Feasible | 15.5 | 16.9 | 18.6 | 20.4 | 28.7 | 0.120 |
| | Δ | −6.5 | −3.7 | −0.9 | 1.6 | 9.4 | 0.172 |

*(Continued)*

| Country | | Bottom 20% | Second 20% | Third 20% | Fourth 20% | Top 20% | Gini index |
|---|---|---|---|---|---|---|---|
| Indonesia | Actual | 7.6 | 11.6 | 15.5 | 21.7 | 43.5 | 0.355 |
| ($\beta^* = 0.017$, $W_{max} = 30.805$) | Feasible | 15.7 | 16.9 | 18.0 | 20.1 | 29.2 | 0.121 |
| | Δ | −8.1 | −5.3 | −2.5 | 1.6 | 14.3 | 0.234 |
| India | Actual | 8.0 | 12.2 | 16.4 | 22.5 | 41.0 | 0.328 |
| ($\beta^* = 0.020$, $W_{max} = 30.302$) | Feasible | 15.3 | 16.7 | 18.1 | 20.4 | 29.4 | 0.128 |
| | Δ | −7.3 | −4.5 | −1.7 | 2.1 | 11.6 | 0.200 |
| Ireland | Actual | 8.9 | 13.1 | 17.0 | 21.9 | 39.2 | 0.301 |
| ($\beta^* = 0.020$, $W_{max} = 29.789$) | Feasible | 15.6 | 17.0 | 18.4 | 20.3 | 28.7 | 0.117 |
| | Δ | −6.7 | −3.9 | −1.4 | 1.6 | 10.5 | 0.184 |
| Iran, Islamic Rep. | Actual | 7.0 | 11.7 | 16.1 | 22.5 | 42.8 | 0.355 |
| ($\beta^* = 0.019$, $W_{max} = 30.107$) | Feasible | 15.1 | 16.5 | 18.0 | 20.3 | 30.1 | 0.135 |
| | Δ | −8.1 | −4.8 | −1.9 | 2.2 | 12.7 | 0.220 |
| Israel | Actual | 5.6 | 11.0 | 16.7 | 23.8 | 43.0 | 0.379 |
| ($\beta^* = 0.022$, $W_{max} = 29.861$) | Feasible | 14.0 | 15.8 | 17.9 | 20.8 | 31.5 | 0.160 |
| | Δ | −8.4 | −4.8 | −1.2 | 3.0 | 11.5 | 0.219 |
| Italy | Actual | 6.5 | 12.1 | 16.9 | 22.9 | 41.5 | 0.348 |
| ($\beta^* = 0.021$, $W_{max} = 29.492$) | Feasible | 14.5 | 16.3 | 18.1 | 20.6 | 30.5 | 0.145 |
| | Δ | −8.0 | −4.2 | −1.2 | 2.3 | 11.0 | 0.203 |
| Jamaica | Actual | 5.7 | 10.3 | 15.2 | 22.7 | 46.1 | 0.402 |
| ($\beta^* = 0.018$, $W_{max} = 30.640$) | Feasible | 14.9 | 16.2 | 17.7 | 20.3 | 30.8 | 0.143 |
| | Δ | −9.2 | −5.9 | −2.5 | 2.4 | 15.3 | 0.259 |
| Kazakhstan | Actual | 9.8 | 13.1 | 16.3 | 21.5 | 39.3 | 0.292 |
| ($\beta^* = 0.018$, $W_{max} = 30.828$) | Feasible | 16.3 | 17.3 | 18.4 | 20.2 | 27.8 | 0.103 |
| | Δ | −6.5 | −4.2 | −2.1 | 1.3 | 11.5 | 0.189 |
| Kenya | Actual | 7.2 | 11.0 | 14.6 | 20.6 | 46.6 | 0.387 |
| ($\beta^* = 0.015$, $W_{max} = 31.130$) | Feasible | 16.1 | 17.0 | 18.0 | 19.7 | 29.3 | 0.117 |
| | Δ | −8.9 | −6.0 | −3.4 | 0.9 | 17.3 | 0.270 |
| Kyrgyz Republic | Actual | 9.5 | 13.3 | 16.8 | 21.9 | 38.5 | 0.288 |
| ($\beta^* = 0.020$, $W_{max} = 30.285$) | Feasible | 15.9 | 17.2 | 18.4 | 20.3 | 28.2 | 0.111 |
| | Δ | −6.4 | −3.9 | −1.6 | 1.6 | 10.3 | 0.177 |
| Korea, Rep. | Actual | 7.5 | 12.4 | 16.9 | 23.2 | 40.0 | 0.329 |
| ($\beta^* = 0.022$, $W_{max} = 30.002$) | Feasible | 14.7 | 16.4 | 18.1 | 20.8 | 30.1 | 0.141 |
| | Δ | −7.2 | −4.0 | −1.2 | 2.4 | 9.9 | 0.188 |
| Lithuania | Actual | 7.0 | 11.7 | 15.6 | 21.6 | 44.1 | 0.367 |
| ($\beta^* = 0.018$, $W_{max} = 30.203$) | Feasible | 15.5 | 16.8 | 18.0 | 20.0 | 29.7 | 0.126 |
| | Δ | −8.5 | −5.1 | −2.4 | 1.6 | 14.4 | 0.241 |
| Luxembourg | Actual | 7.2 | 12.3 | 17.3 | 23.1 | 40.0 | 0.327 |
| ($\beta^* = 0.022$, $W_{max} = 29.726$) | Feasible | 14.5 | 16.3 | 18.2 | 20.7 | 30.2 | 0.143 |
| | Δ | −7.3 | −4.0 | −0.9 | 2.4 | 9.8 | 0.184 |
| Latvia | Actual | 7.1 | 12.1 | 16.6 | 22.7 | 41.5 | 0.343 |
| ($\beta^* = 0.020$, $W_{max} = 29.837$) | Feasible | 14.9 | 16.5 | 18.1 | 20.5 | 30.1 | 0.137 |
| | Δ | −7.8 | −4.4 | −1.5 | 2.2 | 11.4 | 0.206 |
| Moldova | Actual | 10.1 | 14.2 | 17.6 | 22.1 | 36.0 | 0.257 |
| ($\beta^* = 0.022$, $W_{max} = 29.743$) | Feasible | 15.7 | 17.2 | 18.6 | 20.5 | 28.0 | 0.112 |
| | Δ | −5.6 | −3.0 | −1.0 | 1.6 | 8.0 | 0.145 |

*(Continued)*

| Country | | Bottom 20% | Second 20% | Third 20% | Fourth 20% | Top 20% | Gini index |
|---|---|---|---|---|---|---|---|
| Mali | Actual | 7.6 | 11.5 | 15.4 | 21.6 | 43.9 | 0.357 |
| ($\beta^* = 0.017$, $W_{max} = 30.823$) | Feasible | 15.8 | 16.9 | 18.0 | 20.0 | 29.3 | 0.121 |
| | Δ | −8.2 | −5.4 | −2.6 | 1.6 | 14.6 | 0.236 |
| Montenegro | Actual | 6.2 | 12.1 | 17.2 | 24.0 | 40.4 | 0.343 |
| ($\beta^* = 0.024$, $W_{max} = 29.752$) | Feasible | 13.8 | 15.9 | 17.9 | 21.1 | 31.3 | 0.161 |
| | Δ | −7.6 | −3.8 | −0.7 | 2.9 | 9.1 | 0.182 |
| Malaysia | Actual | 5.9 | 10.2 | 15.0 | 22.1 | 46.9 | 0.407 |
| ($\beta^* = 0.017$, $W_{max} = 30.717$) | Feasible | 15.2 | 16.4 | 17.8 | 20.1 | 30.6 | 0.138 |
| | Δ | −9.3 | −6.2 | −2.8 | 2.0 | 16.3 | 0.269 |
| Niger | Actual | 8.7 | 12.4 | 15.9 | 20.8 | 42.2 | 0.329 |
| ($\beta^* = 0.017$, $W_{max} = 30.559$) | Feasible | 16.2 | 17.3 | 18.3 | 19.9 | 28.3 | 0.107 |
| | Δ | −7.5 | −4.9 | −2.4 | 0.9 | 13.9 | 0.222 |
| Netherlands | Actual | 9.4 | 14.6 | 18.3 | 22.5 | 35.2 | 0.257 |
| ($\beta^* = 0.026$, $W_{max} = 28.933$) | Feasible | 14.8 | 16.9 | 18.6 | 20.8 | 28.8 | 0.127 |
| | Δ | −5.4 | −2.3 | −0.3 | 1.7 | 6.4 | 0.130 |
| Panama | Actual | 3.7 | 7.9 | 12.5 | 20.2 | 55.6 | 0.509 |
| ($\beta^* = 0.014$, $W_{max} = 31.651$) | Feasible | 15.4 | 16.3 | 17.4 | 19.3 | 31.5 | 0.141 |
| | Δ | −11.7 | −8.4 | −4.9 | 0.9 | 24.1 | 0.368 |
| Peru | Actual | 5.8 | 10.6 | 15.2 | 22.1 | 46.3 | 0.401 |
| ($\beta^* = 0.017$, $W_{max} = 30.412$) | Feasible | 15.1 | 16.4 | 17.8 | 20.1 | 30.6 | 0.138 |
| | Δ | −9.3 | −5.8 | −2.6 | 2.0 | 15.7 | 0.263 |
| Philippines | Actual | 6.5 | 10.4 | 14.3 | 20.8 | 48.0 | 0.407 |
| ($\beta^* = 0.015$, $W_{max} = 31.200$) | Feasible | 15.9 | 16.8 | 17.8 | 19.7 | 29.8 | 0.123 |
| | Δ | −9.4 | −6.4 | −3.5 | 1.1 | 18.2 | 0.284 |
| Poland | Actual | 8.6 | 13.8 | 17.7 | 22.6 | 37.3 | 0.285 |
| ($\beta^* = 0.024$, $W_{max} = 29.214$) | Feasible | 14.9 | 16.8 | 18.4 | 20.7 | 29.2 | 0.130 |
| | Δ | −6.3 | −3.0 | −0.7 | 1.9 | 8.1 | 0.155 |
| Portugal | Actual | 7.4 | 12.2 | 16.1 | 21.7 | 42.6 | 0.346 |
| ($\beta^* = 0.018$, $W_{max} = 29.901$) | Feasible | 15.4 | 16.9 | 18.1 | 20.1 | 29.5 | 0.125 |
| | Δ | −8.0 | −4.7 | −2.0 | 1.6 | 13.1 | 0.221 |
| Paraguay | Actual | 5.5 | 9.7 | 14.4 | 21.5 | 48.9 | 0.429 |
| ($\beta^* = 0.016$, $W_{max} = 31.002$) | Feasible | 15.4 | 16.4 | 17.7 | 19.8 | 30.6 | 0.135 |
| | Δ | −9.9 | −6.7 | −3.3 | 1.7 | 18.3 | 0.294 |
| Romania | Actual | 6.0 | 12.4 | 17.7 | 24.0 | 39.9 | 0.339 |
| ($\beta^* = 0.025$, $W_{max} = 29.264$) | Feasible | 13.5 | 15.8 | 18.1 | 21.2 | 31.5 | 0.166 |
| | Δ | −7.5 | −3.4 | −0.4 | 2.8 | 8.4 | 0.173 |
| Russian Federation | Actual | 6.9 | 11.8 | 16.2 | 22.7 | 42.4 | 0.351 |
| ($\beta^* = 0.020$, $W_{max} = 30.076$) | Feasible | 15.0 | 16.5 | 18.0 | 20.4 | 30.1 | 0.137 |
| | Δ | −8.1 | −4.7 | −1.8 | 2.3 | 12.3 | 0.214 |
| Senegal | Actual | 7.3 | 11.5 | 15.6 | 21.6 | 43.9 | 0.362 |
| ($\beta^* = 0.017$, $W_{max} = 30.588$) | Feasible | 15.7 | 16.8 | 18.1 | 20.0 | 29.4 | 0.123 |
| | Δ | −8.4 | −5.3 | −2.5 | 1.6 | 14.5 | 0.239 |
| El Salvador | Actual | 5.6 | 10.9 | 15.9 | 22.7 | 45.0 | 0.390 |
| ($\beta^* = 0.019$, $W_{max} = 29.882$) | Feasible | 14.7 | 16.2 | 17.8 | 20.3 | 31.0 | 0.147 |
| | Δ | −9.1 | −5.3 | −1.9 | 2.4 | 14.0 | 0.243 |

*(Continued)*

**Table 3.** (Continued)

| Country | | Bottom 20% | Second 20% | Third 20% | Fourth 20% | Top 20% | Gini index |
|---|---|---|---|---|---|---|---|
| Serbia | Actual | 7.1 | 12.9 | 17.1 | 22.6 | 40.3 | 0.331 |
| ($\beta^*=0.022$, $W_{max}=29.096$) | Feasible | 14.7 | 16.6 | 18.2 | 20.5 | 30.1 | 0.139 |
| | Δ | −7.6 | −3.7 | −1.1 | 2.1 | 10.2 | 0.192 |
| Slovak Republic | Actual | 9.1 | 15.2 | 19.0 | 23.4 | 33.3 | 0.241 |
| ($\beta^*=0.035$, $W_{max}=29.444$) | Feasible | 13.1 | 16.2 | 18.5 | 21.6 | 30.5 | 0.161 |
| | Δ | −4.0 | −1.0 | 0.5 | 1.8 | 2.8 | 0.080 |
| Slovenia | Actual | 10.2 | 14.7 | 18.1 | 22.4 | 34.6 | 0.243 |
| ($\beta^*=0.025$, $W_{max}=29.458$) | Feasible | 15.2 | 17.1 | 18.6 | 20.8 | 28.3 | 0.119 |
| | Δ | −5.0 | −2.4 | −0.5 | 1.6 | 6.3 | 0.124 |
| Sweden | Actual | 7.8 | 13.7 | 17.6 | 23.2 | 37.7 | 0.298 |
| ($\beta^*=0.025$, $W_{max}=29.208$) | Feasible | 14.3 | 16.5 | 18.2 | 20.9 | 30.1 | 0.144 |
| | Δ | −6.5 | −2.8 | −0.6 | 2.3 | 7.6 | 0.154 |
| Togo | Actual | 6.8 | 11.0 | 15.4 | 21.6 | 45.2 | 0.379 |
| ($\beta^*=0.017$, $W_{max}=30.597$) | Feasible | 15.5 | 16.7 | 18.0 | 20.0 | 29.8 | 0.127 |
| | Δ | −8.7 | −5.7 | −2.6 | 1.6 | 15.4 | 0.252 |
| Thailand | Actual | 7.6 | 11.5 | 15.7 | 22.5 | 42.7 | 0.349 |
| ($\beta^*=0.018$, $W_{max}=30.882$) | Feasible | 15.5 | 16.6 | 18.0 | 20.4 | 29.6 | 0.128 |
| | Δ | −7.9 | −5.1 | −2.3 | 2.1 | 13.1 | 0.221 |
| Tonga | Actual | 9.3 | 13.8 | 17.6 | 22.7 | 36.6 | 0.271 |
| ($\beta^*=0.024$, $W_{max}=29.816$) | Feasible | 15.1 | 16.8 | 18.4 | 20.8 | 28.9 | 0.126 |
| | Δ | −5.8 | −3.0 | −0.8 | 1.9 | 7.7 | 0.145 |
| Tunisia | Actual | 7.7 | 12.3 | 16.4 | 21.9 | 41.6 | 0.337 |
| ($\beta^*=0.019$, $W_{max}=29.998$) | Feasible | 15.4 | 16.8 | 18.2 | 20.2 | 29.3 | 0.124 |
| | Δ | −7.7 | −4.5 | −1.8 | 1.7 | 12.3 | 0.213 |
| Türkiye | Actual | 5.2 | 9.5 | 14.2 | 21.0 | 50.1 | 0.444 |
| ($\beta^*=0.015$, $W_{max}=30.961$) | Feasible | 15.4 | 16.5 | 17.7 | 19.7 | 30.7 | 0.135 |
| | Δ | −10.2 | −7.0 | −3.5 | 1.3 | 19.4 | 0.309 |
| Uruguay | Actual | 5.7 | 10.2 | 15.1 | 22.2 | 46.8 | 0.408 |
| ($\beta^*=0.017$, $W_{max}=30.636$) | Feasible | 15.1 | 16.3 | 17.8 | 20.1 | 30.7 | 0.139 |
| | Δ | −9.4 | −6.1 | −2.7 | 2.1 | 16.1 | 0.269 |
| U.S.A. | Actual | 6.1 | 10.7 | 15.3 | 22.2 | 45.7 | 0.397 |
| ($\beta^*=0.018$, $W_{max}=30.508$) | Feasible | 15.2 | 16.4 | 17.8 | 20.1 | 30.4 | 0.137 |
| | Δ | −9.1 | −5.7 | −2.5 | 2.1 | 15.3 | 0.260 |

**Table 4. Descriptive statistics of the optimal income distributions by quintile representing feasible income equality, along with the corresponding values of the Gini index, $\beta^*$, and $W_{max}$ of 71 countries.**

| | Bottom 20% | Second 20% | Third 20% | Fourth 20% | Top 20% | Gini index | $\beta^*$ | $W_{max}$ |
|---|---|---|---|---|---|---|---|---|
| **Mean** | 15.2 | 16.6 | 18.1 | 20.3 | 29.9 | 0.132 | 0.020 | 30.188 |
| **Median** | 15.3 | 16.5 | 18.1 | 20.3 | 29.9 | 0.135 | 0.019 | 30.113 |
| **Mode** | – | – | – | – | – | – | – | – |
| **Standard Deviation** | 0.6 | 0.4 | 0.3 | 0.5 | 1.0 | 0.013 | 0.004 | 0.724 |
| **Minimum** | 13.1 | 15.8 | 17.3 | 19.0 | 27.8 | 0.103 | 0.013 | 28.933 |
| **Maximum** | 16.3 | 17.3 | 18.6 | 21.6 | 31.8 | 0.166 | 0.035 | 32.090 |

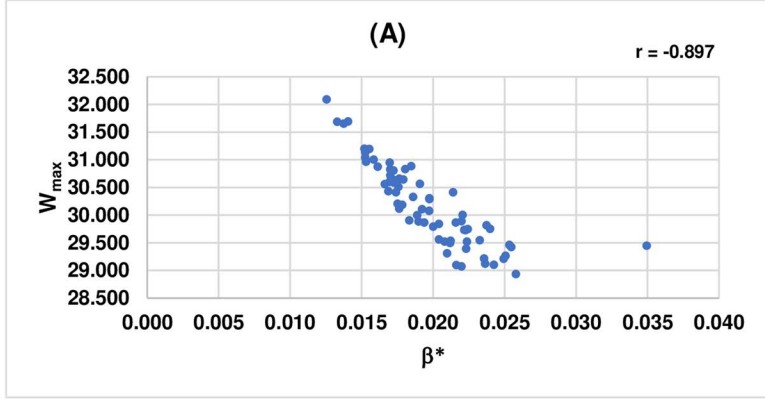

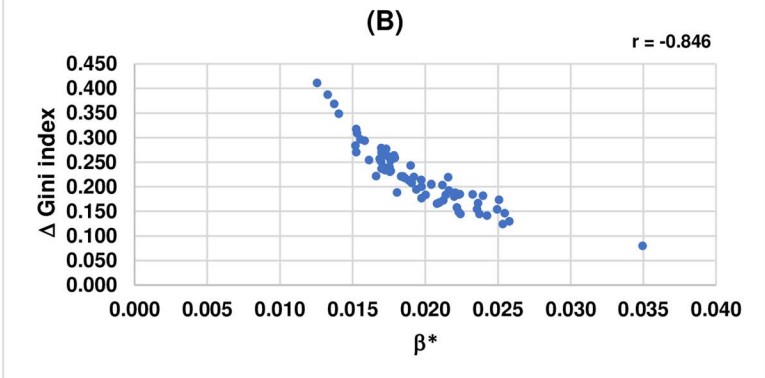

**Fig 3. (A) The correlation between $\beta^*$ and $W_{max}$. (B) The correlation between $\beta^*$ and $\Delta$ Gini index.**

(Czechia, Kazakhstan, and Niger), 18.6% (Moldova, Netherlands, and Slovenia), 21.6% (Slovak Republic), and 31.8% (Colombia). Regarding the Gini index, the minimum value is 0.103 (Kazakhstan) while the maximum value is 0.166 (Romania). In addition, compared to the vast difference in the shapes of the Lorenz curves depicting inequalities in the actual income distributions by quintile of 71 countries, as illustrated in Fig 1, the shapes of the Lorenz curves depicting the optimal income distributions by quintile representing feasible income equality look quite similar across 71 countries, as illustrated in Fig 4.

The results of the similarities in the optimal income distributions by quintile, the values of the Gini index, and the shapes of the Lorenz curves across 71 countries, as shown in Table 3 and Fig 4, confirm Park and Kim [26]'s conjecture in that the universal feasible equality line could be identified and applied across multiple countries.

## Discussion

Park and Kim [26] introduce a method for calculating the optimal income distribution representing feasible income equality by using the sigmoid function and the Boltzmann distribution. This method not only maximizes total social welfare but also provides an unbiased allocation of income among different groups of population in a country. Park and Kim [26] then use the data on quintile income shares and the Gini index of four countries, namely, U.S.A., China, Finland, and South Africa, to demonstrate the concept as a proof of principle, how their method could be used in practice. The results indicate that the optimal quintile income distributions representing feasible income equality, the corresponding values of the Gini index, and the respective shapes of the Lorenz curves of four countries are quite similar to each other. These results lead Park

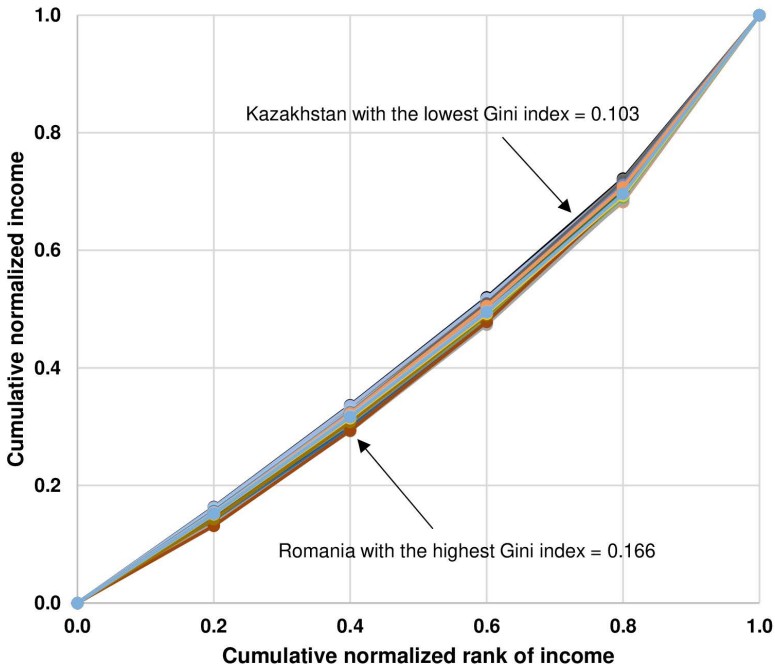

**Fig 4. Lorenz curves depicting the optimal income distributions by quintile representing feasible income equality of 71 countries.**

and Kim [26] to conjecture that there is possibly a universal feasible equality line, as shown by the Lorenz curve, which could be applicable to multiple countries.

This study verifies Park and Kim [26]'s conjecture by employing the data on income shares by quintile and the Gini index of 71 countries in 2021 from the World Bank [35] and using the sigmoid function jointly with the Boltzmann distribution, as specified in Park and Kim [26], in order to calculate the optimal income distributions by quintile representing feasible income equality and maximizing total social welfare for these countries. The overall results confirm Park and Kim [26]'s conjecture in that the optimal income distributions by quintile representing feasible income equality, the corresponding values of the Gini index, and the respective shapes of the Lorenz curves of these 71 countries are not markedly different from each other, suggesting that there is a universal feasible equality line that is applicable to multiple countries. Given that studies, especially in econophysics, have consistently shown that income and wealth distributions exhibit a property of scale invariance or self-similarity in that the shape of income and wealth distributions is statistically stable across space and time [36–41], the results of the optimal income distributions by quintile representing feasible income equality of these 71 countries should not be significantly affected by the choice of period used for studying.

In addition, the difference between actual and optimal income distributions of 71 countries, as shown by Δ quintile income shares and Δ Gini index reported in Table 3 in Results, implies that some countries are closer to, while others are farther below, their optimal income distributions representing feasible income equality. Given that economic and political institutions play a critical role in shaping income distribution of a country, as empirically documented by Acemoglu and Robinson [42], examining the correlations between the quality of economic and political institutions and the difference between actual and optimal income distributions of 71 countries may provide insights into why some countries are closer to, whereas others are well below, their optimal income distributions representing feasible income equality.

In order to examine the correlations between the quality of economic and political institutions and the gap between actual and optimal income distributions representing feasible income equality of 71 countries, this study uses the data on Worldwide Governance Indicators (WGIs) in 2021 from the World Bank [43] as measures of the quality of economic and

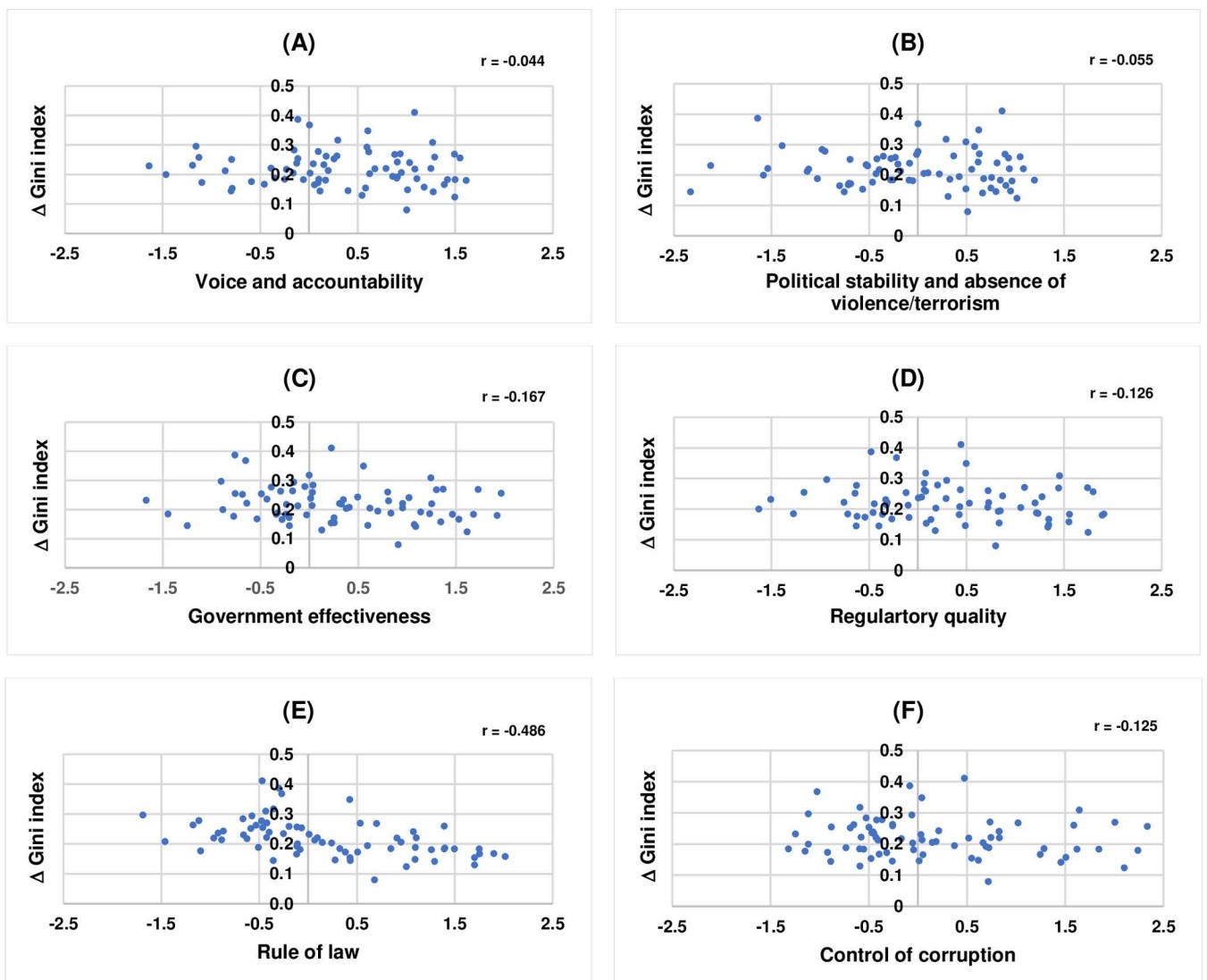

political institutions and the values of Δ Gini index, reported in Table 3 in Results, as a representative for the gap between actual and optimal income distribution. According to the World Bank [43], WGIs comprise six dimensions which are: 1) voice and accountability, 2) political stability and absence of violence/terrorism, 3) government effectiveness, 4) regulatory quality, 5) rule of law, and 6) control of corruption. All indicators take values between −2.5 and 2.5, with higher value corresponding to better institutional quality. The data on WGIs are publicly available and can be accessed from the World Bank [43]. Fig 5 shows the scatter plots of the correlations between each of WGIs and Δ Gini index of 71 countries.

The correlations between each of WGIs and Δ Gini index, as shown in Fig 5, are all negative, with the values of r being equal to −0.044 for voice and accountability, −0.055 for political stability and absence of violence/terrorism, −0.167 for government effectiveness, −0.126 for regulatory quality, −0.486 for rule of law, and −0.125 for control of corruption, respectively. These findings suggest that the better the quality of economic and political institutions is, the closer a

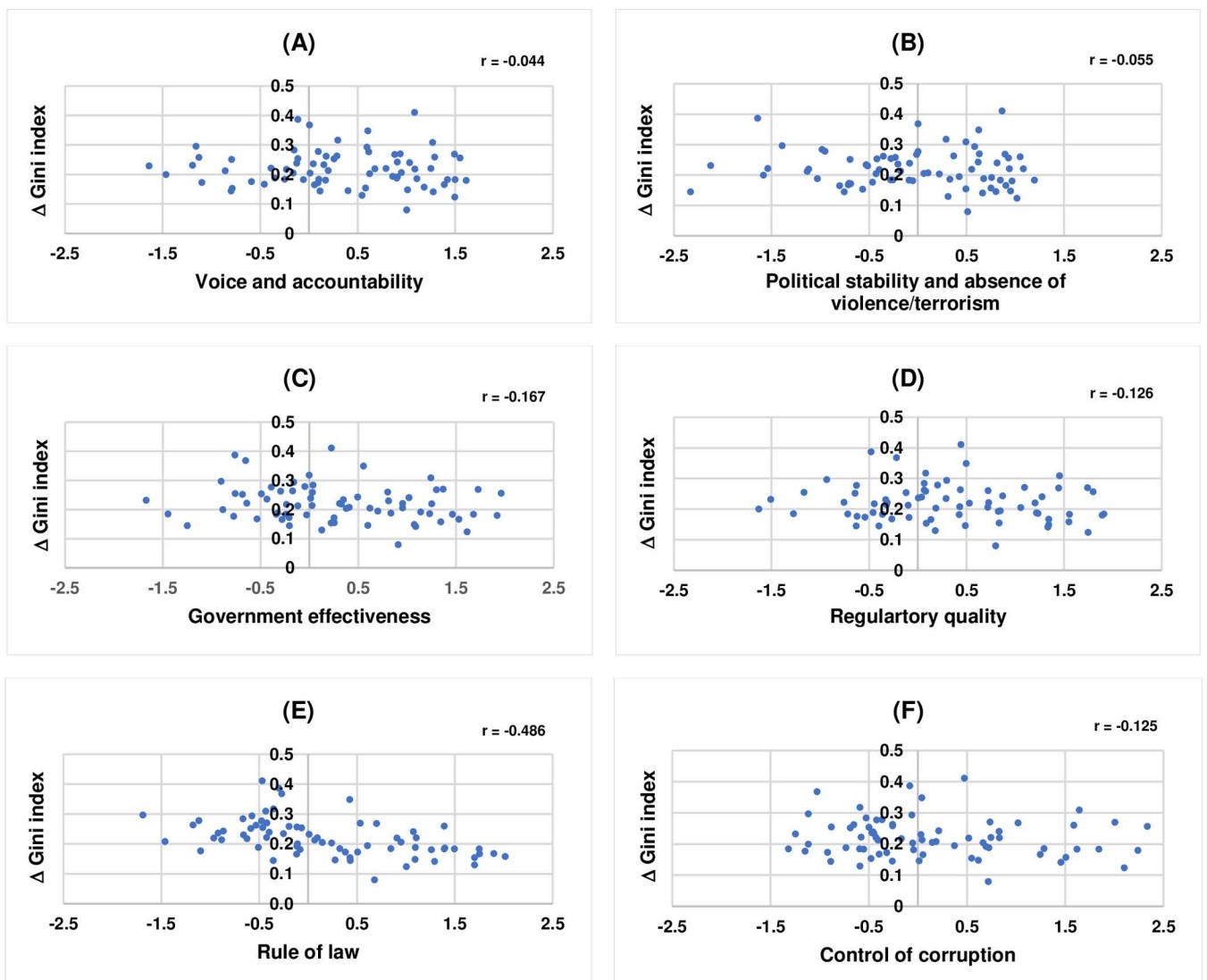

**Fig 5. The correlations between WGIs and Δ Gini index.** (A) Voice and accountability. (B) Political stability and absence of violence/terrorism. (C) Government effectiveness. (D) Regulatory quality. (E) Rule of law. (F) Control of corruption.

country's actual income distribution to the optimal income distribution representing feasible income equality, with the rule of law showing the strongest correlation.

Furthermore, while Park and Kim [26]'s method can be used to calculate the optimal quintile income distribution representing feasible income equality for each country, this study would like to note that the information on the actual quintile income shares and the calculated optimal quintile income shares representing feasible income equality of 71 countries, as reported in Table 3 in Results, can be used to find any approximate level of feasible income share for a particular level of actual income share. This can be done by plotting the Cartesian coordinate where the abscissa represents the actual income share, denoted as $x_i$, and the ordinate represents the feasible income share, denoted as $z_i$. The resulting scatter plot illustrates the relationship between $x_i$s and $z_i$s as shown in Fig 6.

Next, an appropriate parametric functional form is needed in order to perform the curve fitting. Given that the scatter plot depicting the relationship between $x_i$s and $z_i$s, as shown in Fig 6, shows a characteristic S-shape with a step, this study therefore devises the sigmoid step function with four parameters, namely, a, b, c, and d, in order to estimate the relationship between $x_i$s and $z_i$s. These four parameters are used for controlling the curvature so that the estimated sigmoid step function would fit the scatter plot, as illustrated in Fig 6. The sigmoid step function is specified as shown in Eq 7.

$$z_i = a + b * \left[ \frac{\left( \frac{e^{c*(x_i-d)}}{1 + e^{c*(x_i-d)}} \right)}{1 + \left( \frac{e^{c*(x_i-d)}}{1 + e^{c*(x_i-d)}} \right)} \right]$$

(7)

By using the curve fitting technique based on minimizing sum of squared errors, the estimated values of parameters a, b, c, and d for the sigmoid step function, as shown in Eq 8, are equal to 14.400, 34.338, 0.142, and 31.124, respectively.

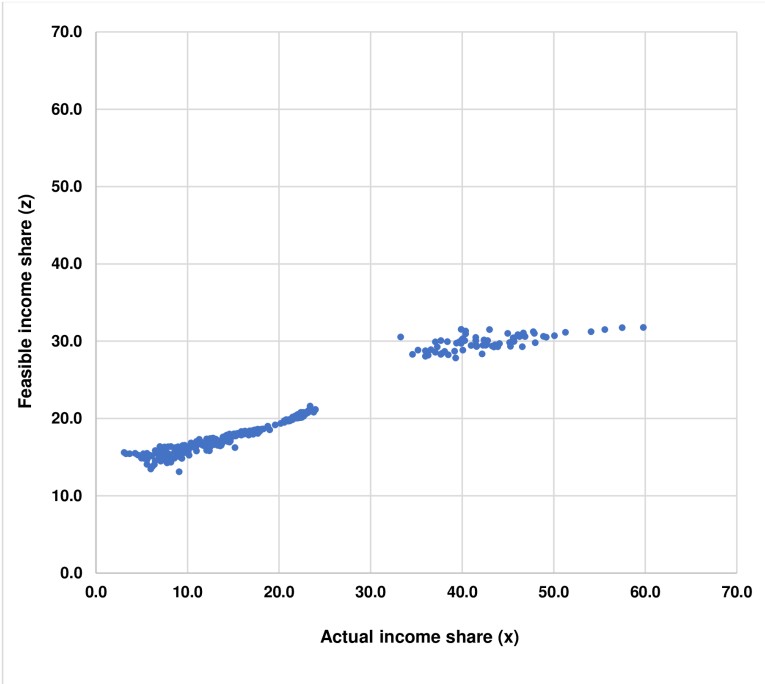

**Fig 6. Scatter plot illustrates the relationship between actual income shares and optimal income shares representing feasible income equality of 71 countries.**

$$z_i = 14.400 + 34.338 * \left[ \frac{\left( \frac{e^{0.142*(x_i-31.124)}}{1+e^{0.142*(x_i-31.124)}} \right)}{1 + \left( \frac{e^{0.142*(x_i-31.124)}}{1+e^{0.142*(x_i-31.124)}} \right)} \right]$$

(8)

Note that the estimated sigmoid step function fits the scatter plot fairly well, with the value of $R^2$ being equal to 0.9874. Fig 7 illustrates the plot of fitted feasible income share for a particular actual income share based on Eq 8.

With the estimated relationship between actual income shares ($x_i$s) and feasible income shares ($z_i$s), as shown in Eq 8 and Fig 7, for a particular level of actual income share, policymakers would be able to find any approximate level of feasible income share which is relatively more convenient than solving Eq 6, as shown in Materials and Methods. These approximate values of feasible income share for a particular value of actual income share could be used as a quantitative benchmark when designing income redistributive policies and measures. For example, if there were no policy intervention, the actual income share would be the same as the feasible income share which can be shown by the 45-degree line, where x = z, as illustrated in Fig 7. In order to redistribute income so that the actual income shares would be closer to the feasible income shares, the income shares of the top 20% and the fourth 20% have to be reduced, with the income share of the top 20% being significantly reduced more than that of the fourth 20% whereas the income shares of the bottom 20%, the second 20%, and the third 20% have to be increased, with the income share of the bottom 20% being increased more than that of the second 20%, and the income share of the second 20% being increased more than that of the third 20% as shown by the arrows in Fig 8.

Given that high income inequality has shown to be associated with health, social, economic, and environmental problems, as discussed in Introduction, and the existing literature has not found unequivocal evidence in favor of income inequality reduction that harms the economy [20], the concept of optimal income distribution representing feasible income equality, originally proposed by Park and Kim [26], could potentially be used as a quantitative benchmark for designing

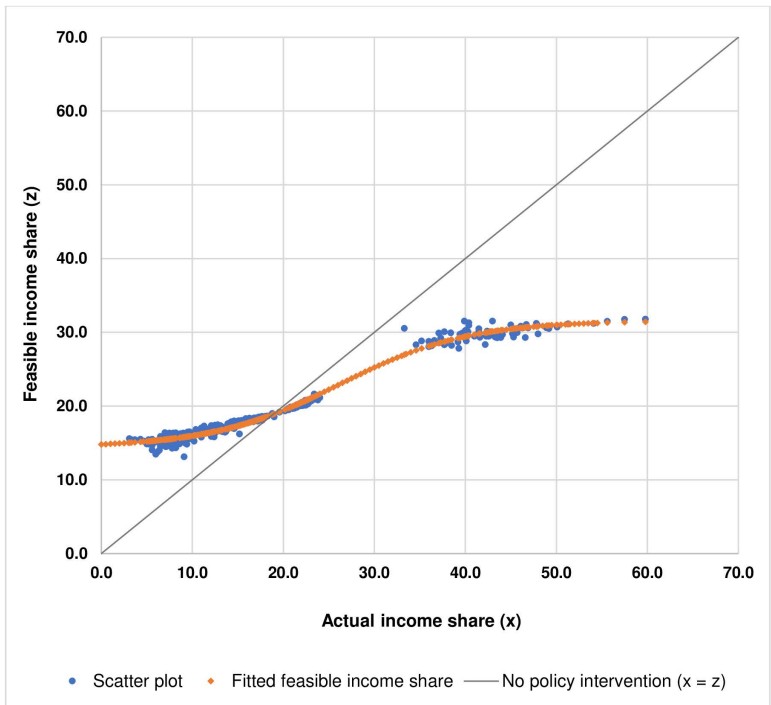

**Fig 7. Fitted plot illustrates the relationship between actual income shares and feasible income shares.**

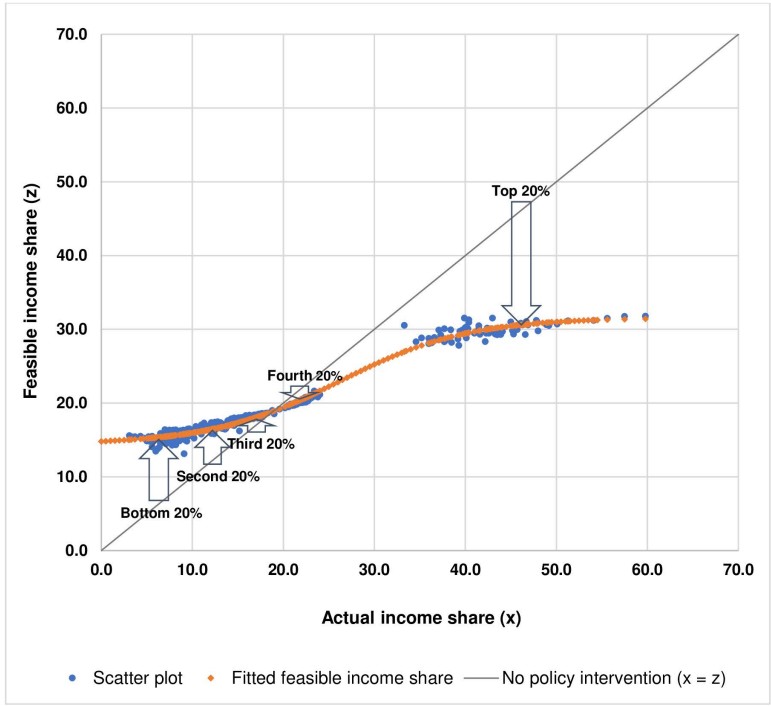

**Fig 8. The arrows demonstrate the use of approximate values of feasible income shares as a quantitative benchmark in income redistributive policy design.**

income redistributive policies and measures. The key challenge is how to formulate and conduct income redistributive policies and measures such that a country or the world as a whole would achieve feasible income equality in reality. The negative correlations between WGIs, especially the rule of law, and $\Delta$ Gini index found in this study point out that the prerequisite, as well suggested by Acemoglu and Robinson [42], is to have economic institutions that enforce property rights, create level of playing field, and encourage investments in new technologies and skills which, in turn, must be supported by political institutions that not only distribute power widely in a pluralistic manner but also are able to achieve some amount of political centralization so as to establish law and order which are the foundations of property rights and well-functioning market economy.

## Supporting information

**S1 Table. The calculated values of L, H, $\mu$, and $\alpha$ of 71 countries.**
(PDF)

## Acknowledgments

The authors sincerely thank Dr. Suradit Holasut and two Reviewers for guidance and comments.

## Author contributions

**Conceptualization:** Thitithep Sitthiyot.

**Formal analysis:** Thitithep Sitthiyot, Kanyarat Holasut.

**Methodology:** Thitithep Sitthiyot, Kanyarat Holasut.

**Validation:** Kanyarat Holasut.

**Writing – original draft:** Thitithep Sitthiyot.

**Writing – review & editing:** Thitithep Sitthiyot, Kanyarat Holasut.

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
