## [Decision Letter · Decision Letter 0]

PONE-D-25-15291A cross-country analysis of feasible income equality using the sigmoid function and the Boltzmann distributionPLOS ONE

Dear Dr. Sitthiyot,

Thank you for submitting your manuscript to PLOS ONE. After careful consideration, we feel that it has merit but does not fully meet PLOS ONE’s publication criteria as it currently stands. Therefore, we invite you to submit a revised version of the manuscript that addresses the points raised during the review process.

The review of your paper is now complete, the Reviewers' reports are below. As you can see, the Reviewers present some important points and a series of recommendations. I also read your paper and concur with these assessments. We kindly ask you to consider all comments and revise the paper accordingly in order to respond fully and in detail to the Reviewers' recommendations.

We look forward to receiving your revised manuscript.

Kind regards,

Pablo Gutierrez Cubillos

Academic Editor

PLOS ONE

Journal Requirements:

2. Your abstract cannot contain citations. Please only include citations in the body text of the manuscript, and ensure that they remain in ascending numerical order on first mention.

Additional Editor Comments :

The review of your paper is now complete, the Reviewers' reports are below. As you can see, the Reviewers present some important points and a series of recommendations. I also read your paper and concur with these assessments. We kindly ask you to consider all comments and revise the paper accordingly in order to respond fully and in detail to the Reviewers' recommendations.

Reviewers' comments:

Reviewer's Responses to Questions

**Comments to the Author**

1. Is the manuscript technically sound, and do the data support the conclusions?

Reviewer #1: Yes

Reviewer #2: Yes

2. Has the statistical analysis been performed appropriately and rigorously? 

Reviewer #1: Yes

Reviewer #2: Yes

3. Have the authors made all data underlying the findings in their manuscript fully available?

Reviewer #1: Yes

Reviewer #2: Yes

4. Is the manuscript presented in an intelligible fashion and written in standard English?

Reviewer #1: Yes

Reviewer #2: Yes

5. Review Comments to the Author

Reviewer #1: This manuscript builds upon the methodological framework developed by Park and Kim (2021), applying the sigmoid utility function and the Boltzmann distribution to estimate optimal income distributions that represent feasible income equality across 71 countries. Using publicly available data from the World Bank, the authors calculate income shares and Gini coefficients for both actual and optimal income distributions, and analyze the extent to which a universal feasible equality line may exist. The study finds a notable convergence across countries in the optimal Gini indices and Lorenz curves, lending support to Park and Kim’s conjecture. The authors also propose an empirical sigmoid step function to approximate feasible income shares from actual income data.

The paper is well-structured, clearly written, and contributes meaningfully to the literature on income distribution and welfare-oriented policy analysis. It confirms and extends a promising theoretical proposal with large-scale empirical evidence. The use of a parametric sigmoid function to estimate feasible income shares from actual shares is a particularly practical contribution. With minor clarifications and improvements in presentation and conceptual framing, the manuscript would merit publication in PLOS ONE.

Below are a few suggestions to enhance the overall quality of the paper:

1. While the authors cite Park and Kim (2021) extensively, it would be helpful to explain more directly why this particular framework was chosen and how the current work builds upon it. A brief discussion in the authors' own words regarding the value and limitations of the original model would better justify the research premise.

2. The sigmoid function adopted from Park and Kim (2021) is rarely used in standard economic modeling. Although the authors present the original justification, their own interpretation or motivation for adopting this functional form would help readers—particularly economists—better follow the argument.

3. It would improve readability if arrows were used to highlight the curves with the lowest and highest Gini index values. Including the corresponding country names and Gini values in the figure or caption would be especially informative.

4. As with Figure 2, please consider using arrows to indicate the curves corresponding to the lowest and highest Gini index values, along with country labels and Gini values.

5. This study uses income data from 2021. Do the authors expect similar findings if data from other years were used? A brief comment on the temporal robustness of the results would be useful, especially considering 2021 may reflect some post-COVID distortions.

6. The curve-fitting function (equations 7 and 8) used to approximate feasible income from actual income is intriguing, but currently underexplained. A short mathematical rationale for the choice of this specific sigmoid step function (including its structure and parameters) would enhance clarity. Relevant references or precedents for this form would also be welcome.

7. The fitted sigmoid function (Equation 8) has potential practical use in policy design. A more explicit discussion of its implications, usability, and limitations in real-world redistributive policymaking would enhance the impact of the study.

Reviewer #2: 1. Overall Assessment

This manuscript builds upon the theoretical framework developed by Park and Kim (2021), applying the sigmoid welfare function and the Boltzmann distribution to estimate optimal income distributions representing feasible income equality across 71 countries. The authors further model the relationship between actual and optimal income shares using a sigmoid step function, which is an interesting empirical extension.

The study represents a meaningful attempt to extend an existing model to a broader cross-country setting and proposes a quantitative benchmark for income inequality evaluation. However, some improvements in clarity, broader literature contextualization, theoretical-to-empirical connection, and presentation would enhance the overall contribution of the manuscript.

2. Major Comments

(1) Clarification of Theoretical Assumptions and Future Research Directions

While the manuscript faithfully applies the Park and Kim (2021) framework, it would benefit from a brief clarification regarding the structure of the contribution factor (Qᵢ). Specifically, the authors could explain why Qᵢ is assumed to be monotonically increasing across quintiles and how it is derived from the income share data. Although a more fundamental redesign of the contribution structure is beyond the scope of this paper, acknowledging this possibility for future work would strengthen the discussion.

(2) Simple Sensitivity Analysis of the β Parameter

Since the β parameter plays a critical role in maximizing total social welfare in this model, a brief sensitivity analysis would improve the robustness of the results. For instance, the authors could provide a simple summary of β values across countries (mean, minimum, maximum) and visualize the relationship between β and ΔGini or the welfare function (W) using a scatter plot. If a full sensitivity analysis is too extensive, a supplementary table or figure would suffice to enhance readers’ understanding.

(3) Suggested Connection to Branko Milanovic’s Work

The manuscript proposes a "universal feasible equality line" based on optimal income distributions across countries. In this regard, it would be valuable to briefly mention Branko Milanovic’s (2016) work, particularly Global Inequality: A New Approach for the Age of Globalization, as a key reference in global income inequality research. Although Milanovic’s "Elephant Curve" analysis differs in structure from the optimization-based approach of the current study, his insights into global inequality dynamics could provide important contextual background. A brief citation and inclusion in the references would enrich the manuscript’s academic depth and international relevance.

(4) Possible Institutional Interpretation: Connecting to Daron Acemoglu's Framework

The manuscript presents differences between actual and optimal income distributions (e.g., ΔGini, Δincome shares) across countries. If the authors could compare these differences with institutional quality indicators (such as World Governance Indicators or Rule of Law Index), it might yield meaningful insights into why some countries are closer to the universal feasible equality line. Such an approach would align with Daron Acemoglu’s political economy theory, which emphasizes the role of institutions in shaping income distribution.

For example:

• A simple cross-country regression or correlation analysis between ΔGini and institutional quality

• Or case studies of contrasting countries (e.g., United States, Sweden, South Africa)

These additions, even at a basic level, could significantly enrich the interpretive depth of the study.

(5) Recommendation for Softening Policy Implication Statements

The manuscript suggests that the proposed model could serve as a practical guideline for policy design. However, given the complex realities of income redistribution involving taxation, welfare systems, and political dynamics, it would be more balanced to present the model as a quantitative benchmark or reference point rather than as a straightforward tool for policy implementation.

3. Conclusion and Recommendation

This manuscript provides a valuable empirical extension of a physics-inspired economic model to a global setting. If the suggested improvements are carefully addressed, the paper will offer a stronger academic contribution as well as more practical relevance for the study of income distribution.

Therefore, I recommend a decision of Major Revision.

6. PLOS authors have the option to publish the peer review history of their article (what does this mean? ). If published, this will include your full peer review and any attached files.

**Do you want your identity to be public for this peer review?** For information about this choice, including consent withdrawal, please see our Privacy Policy .

Reviewer #1: No

Reviewer #2: No

---

## [Author Response · Author response to Decision Letter 1]

13 Jun 2025

Manuscript Number: PONE-D-25-15291R1

Title: A cross-country analysis of feasible income equality using the sigmoid function and the Boltzmann distribution

We sincerely thank Reviewer #1 and Reviewer #2 for providing very useful comments and suggestions. We would like to inform both Reviewers that we have revised our manuscript according to comments and suggestions made by both Reviewers. We hope that our revised manuscript is clearer in all aspects that Reviewer #1 and Reviewer #2 have commented on and suggested. Let us respond to both Reviewers’ comments and suggestions as follows.

Reviewer #1

This manuscript builds upon the methodological framework developed by Park and Kim (2021), applying the sigmoid utility function and the Boltzmann distribution to estimate optimal income distributions that represent feasible income equality across 71 countries. Using publicly available data from the World Bank, the authors calculate income shares and Gini coefficients for both actual and optimal income distributions, and analyze the extent to which a universal feasible equality line may exist. The study finds a notable convergence across countries in the optimal Gini indices and Lorenz curves, lending support to Park and Kim’s conjecture. The authors also propose an empirical sigmoid step function to approximate feasible income shares from actual income data.

The paper is well-structured, clearly written, and contributes meaningfully to the literature on income distribution and welfare-oriented policy analysis. It confirms and extends a promising theoretical proposal with large-scale empirical evidence. The use of a parametric sigmoid function to estimate feasible income shares from actual shares is a particularly practical contribution. With minor clarifications and improvements in presentation and conceptual framing, the manuscript would merit publication in PLOS ONE.

Below are a few suggestions to enhance the overall quality of the paper:

We would like to sincerely thank Reviewer #1 for finding our paper useful.

1. While the authors cite Park and Kim (2021) extensively, it would be helpful to explain more directly why this particular framework was chosen and how the current work builds upon it. A brief discussion in the authors' own words regarding the value and limitations of the original model would better justify the research premise.

In response to Reviewer #1’s comment and suggestion on this issue, in our revised manuscript, we discuss our motivation for choosing Park and Kim (2021)’s method in Introduction, paragraph 2. The contributions of our study are provided in Introduction, paragraph 8 and also in Discussion, paragraph 11. In addition, we state the limitation of Park and Kim (2021)’s method that uses quintile income share as a proxy for income contribution factor which is a measure of economic contribution in Materials and Methods, paragraph 4.

While we cite Park and Kim (2021) extensively, we would like to inform Reviewer #1 that we try our best to explain their concepts and methodologies in our own words. However, we need to give credit to Park and Kim (2021) by citing their work.

2. The sigmoid function adopted from Park and Kim (2021) is rarely used in standard economic modeling. Although the authors present the original justification, their own interpretation or motivation for adopting this functional form would help readers—particularly economists—better follow the argument.

In response to Reviewer #1’s comment and suggestion on this issue, we cite studies in economics that analyze well-being and welfare by using the sigmoid function in Introduction, paragraph 2 and include them in References in our revised manuscript. We also discuss, in our own words, the rationale for using the sigmoid function in Introduction, paragraph 3 in our revised manuscript. Since the sigmoid function is proposed by Park and Kim (2021), we view that it is necessary to acknowledge them by citing their work.

3. It would improve readability if arrows were used to highlight the curves with the lowest and highest Gini index values. Including the corresponding country names and Gini values in the figure or caption would be especially informative.

In our revised manuscript, we add arrows to highlight the Lorenz curves of countries with the lowest and the highest Gini index in Figures 1, 2, and 4 as Reviewer #1 recommends.

4. As with Figure 2, please consider using arrows to indicate the curves corresponding to the lowest and highest Gini index values, along with country labels and Gini values.

We have done this in our revised manuscript for all relevant Figures as stated above.

5. This study uses income data from 2021. Do the authors expect similar findings if data from other years were used? A brief comment on the temporal robustness of the results would be useful, especially considering 2021 may reflect some post-COVID distortions.

In response to Reviewer #1’s concern regarding the period used in our study, we would like to inform Reviewer #1 that we would expect similar findings if the data from other years were used. The reason is that many studies have shown that income and wealth distributions have a property of scale invariance or self-similarity in that the shape of income and wealth distributions, as illustrated by the Lorenz curve, is statistically stable across space and time. Thus, the results of optimal income distributions representing feasible income equality of 71 countries should not be significantly affected by the choice of period used for studying. We explain this in Discussion, paragraph 2 in our revised manuscript. Studies showing that income and wealth distributions have a property of scale invariance or self-similarity are also included in References.

6. The curve-fitting function (equations 7 and 8) used to approximate feasible income from actual income is intriguing, but currently underexplained. A short mathematical rationale for the choice of this specific sigmoid step function (including its structure and parameters) would enhance clarity. Relevant references or precedents for this form would also be welcome.

In response to Reviewer #1’s comment on this issue, we would like to inform Reviewer #1 that we devise this sigmoid step function for the purpose of fitting the scatter plot since the relationship between the actual income shares and the optimal income shares representing feasible income equality has a characteristic S-shape. In addition, the parameters a, b, c, and d are used for controlling the curvature so that the estimated sigmoid step function would fit the scatter plot. We explain this in Discussion, paragraph 7 in our revised manuscript.

7. The fitted sigmoid function (Equation 8) has potential practical use in policy design. A more explicit discussion of its implications, usability, and limitations in real-world redistributive policymaking would enhance the impact of the study.

We would like to thank Reviewer #1 for this very useful suggestion. We provide the discussion regarding the use of the fitted sigmoid step function as a quantitative benchmark for policy design in Discussion, paragraph 10 and Figure 8 in our revised manuscript.

Reviewer #2

This manuscript builds upon the theoretical framework developed by Park and Kim (2021), applying the sigmoid welfare function and the Boltzmann distribution to estimate optimal income distributions representing feasible income equality across 71 countries. The authors further model the relationship between actual and optimal income shares using a sigmoid step function, which is an interesting empirical extension.

The study represents a meaningful attempt to extend an existing model to a broader cross-country setting and proposes a quantitative benchmark for income inequality evaluation. However, some improvements in clarity, broader literature contextualization, theoretical-to-empirical connection, and presentation would enhance the overall contribution of the manuscript.

We sincerely thank Reviewer #2 for finding our work interesting.

2. Major Comments

(1) Clarification of Theoretical Assumptions and Future Research Directions

While the manuscript faithfully applies the Park and Kim (2021) framework, it would benefit from a brief clarification regarding the structure of the contribution factor (Qᵢ). Specifically, the authors could explain why Qᵢ is assumed to be monotonically increasing across quintiles and how it is derived from the income share data. Although a more fundamental redesign of the contribution structure is beyond the scope of this paper, acknowledging this possibility for future work would strengthen the discussion.

In response to Reviewer #2’s comment on this issue, we explain the reason for using income share by quintile as a proxy for income distribution factor which is a measure of economic contribution in Introduction, paragraph 4 and also in Materials and Methods, paragraph 4 in our revised manuscript.

(2) Simple Sensitivity Analysis of the β Parameter

Since the β parameter plays a critical role in maximizing total social welfare in this model, a brief sensitivity analysis would improve the robustness of the results. For instance, the authors could provide a simple summary of β values across countries (mean, minimum, maximum) and visualize the relationship between β and ΔGini or the welfare function (W) using a scatter plot. If a full sensitivity analysis is too extensive, a supplementary table or figure would suffice to enhance readers’ understanding.

As suggested by Reviewer #2, we report the values of β* and Wmax as well as their descriptive statistics in Results, Tables 3 and 4, respectively. A brief discussion about and the scatter plots illustrating the correlation between β* and Wmax and that between β* and ∆ Gini index are provided in Results, paragraph 5 and Figure 3 in our revised manuscript.

(3) Suggested Connection to Branko Milanovic’s Work

The manuscript proposes a "universal feasible equality line" based on optimal income distributions across countries. In this regard, it would be valuable to briefly mention Branko Milanovic’s (2016) work, particularly Global Inequality: A New Approach for the Age of Globalization, as a key reference in global income inequality research. Although Milanovic’s "Elephant Curve" analysis differs in structure from the optimization-based approach of the current study, his insights into global inequality dynamics could provide important contextual background. A brief citation and inclusion in the references would enrich the manuscript’s academic depth and international relevance.

We mention Milanovic’s book entitled “Global inequality: A new approach for the age of globalization” in Introduction, paragraph 1 in our revised manuscript as Reviewer #2 recommends.

(4) Possible Institutional Interpretation: Connecting to Daron Acemoglu's Framework

The manuscript presents differences between actual and optimal income distributions (e.g., Δ Gini, Δ income shares) across countries. If the authors could compare these differences with institutional quality indicators (such as World Governance Indicators or Rule of Law Index), it might yield meaningful insights into why some countries are closer to the universal feasible equality line. Such an approach would align with Daron Acemoglu’s political economy theory, which emphasizes the role of institutions in shaping income distribution.

For example:

• A simple cross-country regression or correlation analysis between Δ Gini and institutional quality

• Or case studies of contrasting countries (e.g., United States, Sweden, South Africa)

These additions, even at a basic level, could significantly enrich the interpretive depth of the study.

We follow Reviewer #2’s suggestion on this point by citing Acemoglu and Robinson (2012)’s work in Discussion, paragraph 3 in our revised manuscript. We then use the data on Worldwide Governance Indicators (WGIs) in 2021 from the World Bank (2024) as a measure of quality of economic and political institutions and the results of Δ Gini index found in our study as a representative for the difference between actual and optimal income distribution to find the correlations between each of WGIs and Δ Gini index. We explain this in Discussion, paragraph 4 in our revised manuscript. Next, we report our results of the correlations between WGIs and Δ Gini index in Discussion, paragraphs 5 and 11 as well as Figure 5 in our revised manuscript. The overall results indicate that there are negative correlations between WGIs and Δ Gini index, implying that the better the quality of economic and political institutions is, the closer a country’s actual income distribution to the optimal income distribution representing feasible income equality.

(5) Recommendation for Softening Policy Implication Statements

The manuscript suggests that the proposed model could serve as a practical guideline for policy design. However, given the complex realities of income redistribution involving taxation, welfare systems, and political dynamics, it would be more balanced to present the model as a quantitative benchmark or reference point rather than as a straightforward tool for policy implementation.

As recommended by Reviewer #2, we soften policy implication statement by stating that the optimal income distributions representing feasible equality could potentially be used as a quantitative benchmark in Abstract, Introduction, paragraph 8, and Discussion, paragraph 11 in our revised manuscript.

3. Conclusion and Recommendation

This manuscript provides a valuable empirical extension of a physics-inspired economic model to a global setting. If the suggested improvements are carefully addressed, the paper will offer a stronger academic contribution as well as more practical relevance for the study of income distribution.

Therefore, I recommend a decision of Major Revision.

We hope that we have made all adjustments according to comments and suggestions made by Reviewer #2.

---

## [Decision Letter · Decision Letter 1]

PONE-D-25-15291R1A cross-country analysis of feasible income equality using the sigmoid function and the Boltzmann distributionPLOS ONE

Dear Dr. Sitthiyot,

Thank you for submitting your manuscript to PLOS ONE. After careful consideration, we feel that it has merit but does not fully meet PLOS ONE’s publication criteria as it currently stands. Therefore, we invite you to submit a revised version of the manuscript that addresses the points raised during the review process.

We look forward to receiving your revised manuscript.

Kind regards,

Pablo Gutierrez Cubillos

Academic Editor

PLOS ONE

Journal Requirements:

Additional Editor Comments (if provided):

Dear Dr. Sitthiyot,

Thank you for submitting the revised version of your manuscript. As noted by the referees, this version represents a marked improvement over the original submission.

To make the paper suitable for publication, I kindly ask that you carefully address all the remaining minor comments raised by the referees. Additionally, please ensure a thorough proofreading of the manuscript, as there are still several punctuation and typographical errors that need to be corrected.

We look forward to receiving your revised manuscript.

Best regards,

Reviewers' comments:

Reviewer's Responses to Questions

**Comments to the Author**

1. If the authors have adequately addressed your comments raised in a previous round of review and you feel that this manuscript is now acceptable for publication, you may indicate that here to bypass the “Comments to the Author” section, enter your conflict of interest statement in the “Confidential to Editor” section, and submit your "Accept" recommendation.

Reviewer #1: All comments have been addressed

Reviewer #2: All comments have been addressed

2. Is the manuscript technically sound, and do the data support the conclusions?

Reviewer #1: Yes

Reviewer #2: Yes

3. Has the statistical analysis been performed appropriately and rigorously? 

Reviewer #1: Yes

Reviewer #2: Yes

4. Have the authors made all data underlying the findings in their manuscript fully available?

Reviewer #1: Yes

Reviewer #2: Yes

5. Is the manuscript presented in an intelligible fashion and written in standard English?

Reviewer #1: Yes

Reviewer #2: Yes

6. Review Comments to the Author

Reviewer #1: The authors have appropriately addressed major concerns raised in the initial round of review. I find the revised manuscript improved and recommend acceptance after minor polishing.

The following sentence in the manuscript reads somewhat awkwardly due to its length and structure:

Given that the sigmoid function is used in well-being and welfare analysis, for example, Hänsel and Quaas [27], Takano et al. [28], and Mancini and Vecchi [29] and the Boltzmann distribution is employed to study income and wealth distributions, for example, Drăgulescu and Yakovenko [30], Pal and Pal [31], Pareschi and Toscani [32], Tao [33], and Ludwig and Yakovenko [34], Park and Kim [26] propose that the sigmoid function can be used jointly with the Boltzmann distribution in order to calculate the optimal income distribution that represents feasible income equality and maximizes total social welfare simultaneously.

I suggest revising it for clarity and natural flow. One possible revision is:

Given that the sigmoid function has been used in well-being and welfare analysis (e.g., Hänsel and Quaas [27]; Takano et al. [28]; Mancini and Vecchi [29]) and that the Boltzmann distribution has been applied to the study of income and wealth distributions (e.g., Drăgulescu and Yakovenko [30]; Pal and Pal [31]; Pareschi and Toscani [32]; Tao [33]; Ludwig and Yakovenko [34]), Park and Kim [26] propose using both the sigmoid function and the Boltzmann distribution to calculate the optimal income distribution that represents feasible income equality and maximizes total social welfare.

Reviewer #2: The manuscript provides robust empirical evidence that the gap between actual and optimal income distributions across 71 countries is closely related to institutional quality, especially the rule of law. The authors also present a clear functional relationship—using a sigmoid function—that enables the estimation of feasible income shares from actual income shares, offering a practical quantitative benchmark for policy design. Policy implications and real-world limitations are discussed in a balanced and convincing manner. Furthermore, the authors have responded diligently and thoroughly to all reviewer comments.

For minor revision, I recommend that the authors simply clarify in the Abstract that “feasible income equality” refers to the ‘optimal income distribution that is realistically attainable.’ No further substantial changes are required.

Overall, I find the manuscript to be of high quality and suitable for publication after this minor clarification.

7. PLOS authors have the option to publish the peer review history of their article (what does this mean? ). If published, this will include your full peer review and any attached files.

**Do you want your identity to be public for this peer review?** For information about this choice, including consent withdrawal, please see our Privacy Policy .

Reviewer #1: No

Reviewer #2: No

---

## [Author Response · Author response to Decision Letter 2]

17 Jul 2025

Manuscript Number: PONE-D-25-15291R2

Title: A cross-country analysis of feasible income equality using the sigmoid function and the Boltzmann distribution

We sincerely thank Reviewer #1 and Reviewer #2 for recommending our revised manuscript for publication in PLOS ONE on the condition that minor adjustments are made. We would like to inform both Reviewers that we have revised our manuscript according to suggestions made by both Reviewers. They are as follows.

Reviewer #1:

The authors have appropriately addressed major concerns raised in the initial round of review. I find the revised manuscript improved and recommend acceptance after minor polishing.

The following sentence in the manuscript reads somewhat awkwardly due to its length and structure:

Given that the sigmoid function is used in well-being and welfare analysis, for example, Hänsel and Quaas [27], Takano et al. [28], and Mancini and Vecchi [29] and the Boltzmann distribution is employed to study income and wealth distributions, for example, Drăgulescu and Yakovenko [30], Pal and Pal [31], Pareschi and Toscani [32], Tao [33], and Ludwig and Yakovenko [34], Park and Kim [26] propose that the sigmoid function can be used jointly with the Boltzmann distribution in order to calculate the optimal income distribution that represents feasible income equality and maximizes total social welfare simultaneously.

I suggest revising it for clarity and natural flow. One possible revision is:

Given that the sigmoid function has been used in well-being and welfare analysis (e.g., Hänsel and Quaas [27]; Takano et al. [28]; Mancini and Vecchi [29]) and that the Boltzmann distribution has been applied to the study of income and wealth distributions (e.g., Drăgulescu and Yakovenko [30]; Pal and Pal [31]; Pareschi and Toscani [32]; Tao [33]; Ludwig and Yakovenko [34]), Park and Kim [26] propose using both the sigmoid function and the Boltzmann distribution to calculate the optimal income distribution that represents feasible income equality and maximizes total social welfare.

We thank Reviewer #1 for this suggestion. We adjust the sentence that Reviewer #1 kindly rewrites it for us so that it is in line with PLOS ONE’s citation style in Introduction, paragraph 2 in our revised manuscript. It reads as follows.

Given that the sigmoid function has been used in well-being and welfare analysis [27-29] and that the Boltzmann distribution has been applied to the study of income and wealth distributions [30-34], Park and Kim [26] propose using both the sigmoid function and the Boltzmann distribution to calculate the optimal income distribution that represents feasible income equality and maximizes total social welfare.

Reviewer #2:

The manuscript provides robust empirical evidence that the gap between actual and optimal income distributions across 71 countries is closely related to institutional quality, especially the rule of law. The authors also present a clear functional relationship—using a sigmoid function—that enables the estimation of feasible income shares from actual income shares, offering a practical quantitative benchmark for policy design. Policy implications and real-world limitations are discussed in a balanced and convincing manner. Furthermore, the authors have responded diligently and thoroughly to all reviewer comments.

For minor revision, I recommend that the authors simply clarify in the Abstract that “feasible income equality” refers to the ‘optimal income distribution that is realistically attainable.’ No further substantial changes are required.

Overall, I find the manuscript to be of high quality and suitable for publication after this minor clarification.

We thank Reviewer #2 for this useful recommendation. We add the sentence that Reviewer #2 suggests in the Abstract in our revised manuscript.

---

## [Editor Report · Decision Letter 2]

A cross-country analysis of feasible income equality using the sigmoid function and the Boltzmann distribution

PONE-D-25-15291R2

Dear Dr. Sitthiyot,

We’re pleased to inform you that your manuscript has been judged scientifically suitable for publication and will be formally accepted for publication once it meets all outstanding technical requirements.

Kind regards,

Pablo Gutierrez Cubillos

Academic Editor

PLOS ONE